

**Annual variability and regulation of methane and sulfate fluxes in**
**Baltic Sea estuarine sediments**
Joanna E. Sawicka and Volker Brüchert
Department of Geological Sciences, Stockholm University, Stockholm, 10691, Sweden
*Correspondence to:* Volker Brüchert (volker.bruchert@geo.su.se)



**Abstract.** The effects of temperature, changes in benthic oxygen concentration, and historical

eutrophication on sediment methane concentrations and benthic fluxes were investigated at two type

localities for open-water coastal and eutrophic, estuarine sediment in the Baltic Sea. Benthic fluxes of

methane and oxygen, sediment porewater concentrations of dissolved sulfate, methane, and $^{35}$S-sulfate

reduction rates were obtained over a 12-month period from April 2012 to April 2013. Benthic methane

fluxes varied by factors of 5 and 12 at the offshore coastal site and the eutrophic estuarine station,

respectively, ranging from 0.1 mmol m$^{-2}$d$^{-1}$ in winter at an open coastal site to 2.6 mmol m$^{-2}$d$^{-1}$ in late

summer in the inner eutrophic estuary. Total oxygen uptake (TOU) and $^{35}$S-sulfate reduction rates

(SRR) correlated with methane fluxes showing low rates in the winter and high rates in the summer.

The highest porewater methane concentrations also varied by factors of 6 and 10 over the sampling

period with lowest values in the winter and highest values in late summer-early autumn. The highest

porewater methane concentrations exceeded 6 mM a few centimeters below the sediment surface, but

never exceeded the in-situ saturation concentration. 21 – 24% of the total sulfate reduction was coupled

to anaerobic methane oxidation lowering methane concentrations below the sediment surface far below

the saturation concentration. These data imply that bubble emission likely plays no or only a minor role

for methane emissions in these sediments. The changes in porewater methane concentrations over the

observation period are too large to be explained by temporal changes in methane formation and

methane oxidation rates. Instead, it appears that advective methane recharge supplies of methane from

deeper sediment layers to near-surface sediment. These are possible related to the transport of methane

from deeper gas-rich areas or due to free gas movement or groundwater discharge.

**Keywords** Methane, sulfate reduction, estuary



## 1 Introduction

The world's estuaries have been estimated to emit between 1.8 and 6.6 Tg $CH_4$ $y^{-1}$ to the atmosphere (Borges and Abril, 2011; Amouroux et al 2002, Marty et al., 2001; Middelburg et al., 2002; Sansone et al., 1999; Upstill-Goddard et al., 2000). As other globally upscaled estimates of emissions, these estimates also have considerable uncertainties. In the case of estuaries, a major cause of the uncertainty are relatively few spatially and temporally resolved measurements of anaerobic carbon degradation measurements in sediments and measurements of methane fluxes from sediments. In estuarine waters methane is produced by methanogenesis in underlying anoxic sediments, lateral freshwater or sewage discharge, seepage of methane-rich groundwater, or transport in the near-shore by aquatic plants (Borges and Abril, 2011). The amount of sedimentary methane production in estuaries is a function of organic matter availability, bottom water oxygen concentrations, and the salinity of the estuary. Methane production is generally greater in low-salinity estuaries because of lower sulfate availability to promote bacterial sulfate reduction (Borges and Abril, 2011). Methane fluxes from estuarine sediments are characterized by significant spatial and temporal variability (Borges and Abril 2011). Temporal patterns show that concentrations and fluxes of $CH_4$ are generally higher in the warmer summer season and low in the colder season (Crill et al., 1983, Martens and Klump, 1984, Musenze et al., 2014; Reindl and Bolałek, 2014). Notably, very few studies have considered $CH_4$ fluxes during snow- and ice-covered periods. While shallow systems within the tidal range derive a significant amount of the methane flux from ebullition (Martens and Klump, 1984), groundwater discharge, tidal pumping, and transport by aquatic plants (Middelburg et al., 2002; Kristensen et al 2008), the transport from deeper systems such as fjords and fjärds is thought to occur largely by molecular diffusion (Abril and Iversen, 2002, Sansone et al., 1998).

Globally more than 90% of methane produced in marine sediments is estimated to be oxidized by the anaerobic oxidation of methane (AOM), mostly in the sulfate-methane transition zone (Knittel and Boetius, 2009, Martens and Berner, 1974; Jørgensen and Parkes, 2010). It is not known how much





methane is oxidized by AOM in estuarine sediments. In addition, up to 90% of the remaining methane
that reaches the sediment surface may be oxidized aerobically at the sediment surface or in the water
column (Reeburgh, 2007). Yet, methane concentrations in estuarine waters are almost always higher
than the atmospheric equilibrium concentration indicating that microbial oxidation processes and
physical exchange with the atmosphere in estuaries are relatively inefficient in removing methane.
Despite its obvious importance, only few studies have specifically addressed anaerobic oxidation of
methane by sulfate and aerobic oxidation in estuarine environments (e.g., Treude et al., 2005, Thang et
al., 2013). The aim of this study was to further elucidate mechanisms behind temporal variability of
methane fluxes in a high-latitude coastal and estuarine environment with winter ice cover. We
determined porewater concentrations of methane and sulfate, measured sulfate reduction rates with the
$^{35}$S-sulfate tracer method, and conducted core incubations to determine benthic fluxes of methane and
oxygen at two deep stations of a low-salinity Baltic Sea estuary inside and at the opening of the estuary
to the Baltic. Investigations were carried out over four seasons to capture the annual variability of
chemical and biological conditions at the sediment surface and their influence on methane dynamics.

**2 Materials and methods**
**2.1 Site description**
Himmerfjärden (Figure 1) is a fjord-type estuary with a surface area of 174 km$^2$ and a N-S
salinity gradient increasing from 5.5‰ in the inner part to 7.0‰ at the opening to the Baltic. It is
morphologically characterized by four basins, divided by sills. Water discharge to the estuary is low
(flushing rate 0.025/day) and derives from land run-off and precipitation (33% and 14% respectively),
outflow from Lake Mälaren in the north (46%), discharge of a major sewage treatment plant (STP)
(7%) (Boesch et al., 2006; Engqvist, 1996). The STP, built in the early 1970s, treats sewage water from
300,000 inhabitants of the southern Stockholm metropolitan area, and its inorganic effluent is
discharged mainly in the form of inorganic nitrogen and phosphorus to the inner basins (Savage and



Elmgren, 2010). The estuary undergoes thermohaline stratification during late summer and autumn, especially in the inner part, which experiences regular seasonal bottom water hypoxia. The tidal range is low (few cm) and relatively cold bottom waters (1.5 - 9°C) dominate throughout the year. Late-summer-early fall bottom water hypoxia has also been reported occasionally for the outer basins of the estuary when winds are weak and circulation is inhibited (Elmgren and Larsson, 1997).

Bottom water and sediment samples were taken from a station in the inner part of Himmerfjärden, Station H6, and from a station located outside the estuary, Station B1 (Figure 1). Samples were collected in April, August, October 2012, and in February 2013. In addition, in April 2013 whole-core incubations were performed to determine methane and oxygen fluxes to record a full year of seasonal variability. Station B1 has soft, olive grey, muddy sediment with a rusty brown surface layer, while the sediment at station H6 is soft, laminated black mud with a 1-2 mm thin brown surface layer that occurs during the winter and spring. Sediment accumulation rates range from 0.98 cm yr$^{-1}$ in the innermost part of the estuary to 0.77 cm yr$^{-1}$ in the outer part of the estuary (Thang et al., 2013).

**2.2 Sample collection**

Sediments with well-preserved sediment surfaces were collected with a Multicorer in acrylic tubes (9.5 cm diameter) to 40 cm depth to determine $^{35}$S-sulfate reduction rates, porosity, and the porewater constituents methane and sulfate. Additional cores were collected for sediment core incubations. Porewater methane samples were immediately collected on-board from the cores as described below. The other cores were capped with rubber stoppers, transported to the marine laboratory on the island of Askö within 90 minutes and kept cold at bottom water temperatures for later experiments and subsampling. In February 2013, ice partially covered Station B1 and there was full ice coverage at Station H6, and sampling was only possible after ice breaking. For whole-core incubations, 30 l of bottom water was collected with a 5 liter HydroBios bottle and kept cold until for the experiments. Temperature, salinity, and oxygen concentrations were determined with a handheld WTW Oxygen meter directly in the water overlying the sediment cores.

130

### 2.3 Organic carbon concentrations and porosity

Surface sediment concentrations of organic carbon were determined on freeze-dried sediment with a

Fisons CHN elemental analyzer after treatment of freeze-dried sediment with 1N HCl to remove

inorganic carbon. Water content (%) was determined by drying 5 ml of sediment at 105°C for two

hours and calculating the percent loss after drying.

136

### 2.4 Methane analysis

Samples for methane were collected directly through the side of taped, pre-drilled core liners and taken

in 2-cm intervals seconds after the core was retrieved on deck. The core sampling method used in this

study permits complete sampling and preservation of porewater methane within 5 minutes after the

core was on deck. Under these circumstances, loss of methane due to gas loss is low and methane

concentrations could be determined for porewaters that were far above the saturation limit at 1

atmosphere pressure for the salinity and temperature range of the bottom water (between 1.9 mM and

2.4 mM). A sediment sample of exactly 2.5 mL was taken with a 3 mL cutoff syringe. The sample was

transferred to a 20 mL serum vial containing 5 mL 5 M NaCl and immediately closed with a thick

septum and an aluminum crimp seal. The sample was shaken, left for 1 hour for gas equilibration, and 5

mL of brine was injected into a sample vial to force out the 5 mL gas samples out of a vial into the

syringe. The $CH_4$ measurements were carried out on a gas chromatograph (GC) with a flame ionization

detector (FID) (SRI 8610C) and $N_2$ was used as carrier gas. $CH_4$ standards 100 ppm and 10000 ppm

(Air Liquide) were used for calibration.

The concentration of methane (mM) in the headspace of a sample was calculated from:

$$CH_4(mM) = \frac{V_{headspeace} A \alpha}{24.1 V_{sediment} \rho} \qquad (1)$$






where $V_{head}$ is the volume of the headspace in the sample vial ($cm^3$), $\rho$ is the sediment porosity, A is the
peak area of methane eluted, $\alpha$ is the slope of the standard curve (parts per million volume basis), and
$V_{sed}$ is the volume of the sediment sample (cubic centimeter). The molar volume of methane at 20 °C
and 1 atm pressure (24.148 L $mol^{-1}$) was used to convert from partial volume of $CH_4$ gas to the mole
fraction of $CH_4$.

**2.5 Sulfate concentration**
Porewater samples for sulfate concentration measurements were obtained using rhizones (Atlas
Copco Welltech) (Seeberg-Elverfeldt et al 2005). Rhizones were treated for 2 hours in 10% HCl
solution, followed by two rinses with deionized water for 2 hours and final storage in deionized water.
Rhizones were connected to 10 mL disposable plastic syringes via a 3-way luer-type stop-cock and
inserted in 1 cm intervals through tight-fitting, pre-drilled holes in the liner of the sediment cores. The
first mL of pore water was discarded from the syringe. No more than 2 ml were collected from each
core to prevent cross-contamination of adjacent due to the porewater suction (Seeberg-Elverfeldt et al.,
2005). Sulfate concentration was measured on a Dionex System IC 20 ion chromatograph.

**2.6 $^{35}$S-Sulfate reduction rates**
To determine bacterial sulfate reduction rates (SRR) sediment cores were subsampled in 40-cm
long 28 mm-diameter cores with 1-cm spaced, silicon-sealed, pre-drilled small holes on the side for
injections. For the incubation, the whole-core incubation method by Jørgensen (1978) was used. $^{35}SO_4^{2-}$
tracer solution was diluted in a 6 ‰ NaCl solution containing 0.5 mM $SO_4^{2-}$ and 2.5 µl of the tracer
solution (50kBq) was injected through the pre-drilled holes. The cores were then capped and sealed in
plastic wrap foil and incubated for 8 hours at the respective bottom water temperatures. After this time,
the incubations were stopped by sectioning the core in 1-cm intervals to 5 cm depth and in two
centimeter intervals below this depth to the bottom of the core. Sediment sections were transferred into
50 ml plastic centrifuge tubes containing 20 ml zinc acetate (20% v/v) and shaken vigorously and





frozen. The total amount of [35]S-labeled reduced inorganic sulfur (TRS) was determined using the
single-step cold distillation method by Kallmeyer et al. (2004). TRS and supernatant sulfate were
counted on a TriCarb 2095 Perkin Elmer scintillation counter. The sulfate reduction rate was calculated
using the following equation (Jørgensen, 1978):
$$^{35}SRR = \left. TRI^{35}S \; \rho \; 1.06 \middle/ (^{35}SO_4^{2-} + TRI^{35}S) \; T \right. \tag{2}$$

where ($SO_4^{2-}$ $\rho$) is the pore water sulfate concentration corrected for porosity $\rho$, $TRI^{35}S$ and $^{35}SO_4^{2-}$ are
the measured counts (cpm) of sulfate and total reduced sulfur species, respectively, 1.06 is a correction
factor accounting for the isotope discrimination of [35]S against [32]S-sulfate, and T is the incubation time.
The sulfate reduction rate is reported as nmol cm$^{-3}$ day$^{-1}$. $^{35}SRR$ were measured in three parallels cores
for all depth intervals and the values reported here are the median values of the triplicates. The
detection limit of the rate measurements accounting for distillation blanks and radioactive decay of [35]S
between experiment and laboratory workup was 0.1 nmol day$^{-1}$ cm$^{-3}$.

## 2.7 Whole-core sediment incubations

Four intact cores with undisturbed sediment surfaces and clear overlying water were subsampled in the
laboratory in acrylic tubes (i.d. 6.2 cm, height 25 cm) retaining about 10 cm of the overlying water. The
sediment height in the tubes was approximately 10 cm. The cores were incubated in a 40-liter
incubation tank filled with bottom water from the same station. Before the incubation the overlying
water in the cores was equilibrated with bottom water in the tank. The overlying water in the cores was
stirred by small magnetic bars mounted in the core liners and driven by an external magnet at 60 rpm.
The cores were pre-incubated uncapped for 6 hours and subsequently capped and incubated for a period
of 6 to 12 hours depending on the initial oxygen concentration in the bottom water.





### 2.8 Total oxygen uptake

Oxygen sensor spots (Firesting oxygen optode, PyroScience GmbH, Germany) with a sensing surface

of a diameter of 5 mm were attached to the inner wall of two incubation cores (diameter 5.5 cm). The

sensor spots were calibrated against $O_2$-saturated bottom water and oxygen-free water following the

manufacturer's guidelines accounting for temperature and salinity of the incubation water.

Measurements were performed with a fiberoptic cable connected to a spot adapter fixed at the outer

core liner wall at the spot position. The $O_2$ concentration was continuously logged during incubations.

Sediment total oxygen uptake (TOU) rates were computed by linear regression of the $O_2$ concentration

over time.

### 2.9 Methane fluxes

Methane fluxes were determined from discrete water samples collected in 12 mL Exetainers (Labco,

Wycombe, UK) prefilled with 50 µL of 50% $ZnCl_2$ without headspace. Samples were collected at the

beginning (time zero) and at the end of the incubation (time final), usually after 24 hours. $CH_4$

concentrations were determined using the headspace equilibration technique (Kampbell et al., 1989) by

displacing 3 ml of the water in the exetainers with high-purity helium gas at atmospheric pressure. The

Exetainers were shaken at 400rpm on a shaking table for 60 minutes to allow the gas to equilibrate

between the headspace and the liquid phase and left to rest for half an hour. After equilibration 2.5 mL

of NaCl brine was injected into an Exetainer to force the gas samples into an injection syringe while

maintaining the headspace pressure. The samples were injected onto a 1 ml injection loop of a gas

chromatograph (SRI 8610) with FID detector using $N_2$ as carrier gas. $CH_4$ standards 5 ppm, 100 ppm

and 10000 ppm (Air Liquide) were used to construct a calibration curve.

Partial pressure of $CH_4$ in the equilibrated headspace and water was calculated using the solubility

coefficient β for $CH_4$ (Wilhelm et al 1977), gas constant R (8.314 L kPa mol$^{-1}$ K$^{-1}$), air pressure (P in



kPa), headspace gas concentration $CH_{4\ (hsp)}$ (ppm), headspace volume (0.003L), water volume in the
exetainer (0.009L), and laboratory temperature (293 K) according to
$CH_4$ (nM) = ($CH_{4\ (hsp)}$ + β $CH_{4\ hsp}$)* P/RT                              (3)
Fluxes (J) of $CH_4$ (mmol $m^{-2}$ $d^{-1}$) during the whole core sediment incubations were calculated according
to
J = ($CH_{4\ start}$ − $CH_{4\ end}$)/T * V/A                              (4)
where $CH_{4\ start}$ and $CH_{4\ final}$ represent the end and start concentrations in mmol/$m^3$, V is headspace
volume (L), A is the surface area of the incubation core ($m^2$), and T is the incubation time (days).

**2.10 Diffusive flux calculations**
Diffusive fluxes of methane and sulfate were estimated from the porewater gradients of methane and
sulfate for the sediment surface and the sulfate-methane transition zone. Sediment cores at station B1
showed occasional burrows from deposit feeders in the topmost 2 cm of sediment, whereas sediment at
station H6 was largely devoid of macro- and meiofauna. Since only one sample was taken from the
topmost 2 cm, quantitative depth-related effects of bioturbation cannot be accounted for in this analysis
and upward diffusive transport of methane was assumed as the dominant transport pathway. Fluxes
were estimated using Fick's first law of diffusion
$J = D_s \frac{dC}{dx}$                              (5)
assuming that flux was dominated by molecular diffusion, where dC is the change in concentration of
dissolved sulfate (mM) or methane (mM) over a depth interval dx (cm), and $D_s$ is the sediment
diffusion coefficient corrected for temperature and salinity according to Boudreau (1996). $D_s$ was
recalculated from the molecular diffusion coefficient $D_o$ for sulfate and methane according to Iversen
and Jørgensen (1994).





## 3 Results

### 3.1 Bottom water temperature, dissolved oxygen, organic carbon

During the observation period April 2012 through February 2013 salinity varied between 5.4 and 6.5‰ at Station H6 and 6.5 and 7.0‰ at station B1 (Table 1), while bottom water temperatures ranged from 2.4°C to 6.9°C at station B1 and 1.8°C to 9.4°C at station H6. The lowest and highest bottom water oxygen concentrations were measured in April 2012 (40 µM at station H6, and 160 µM at station B1) and February 2013 (300 at station H6 and 380 µM at station B1, respectively). Surface sediment organic carbon concentrations were similar at the two stations ranging between 4.6 and 5.2% at Station B1, and 5.0% and 6.0% at Station H6.

### 3.2 Methane and sulfate concentrations

The highest methane concentrations in the sediment cores were recorded in August, when they reached 5.7 mM at station H6 and 1.9 mM at station B1 (Figure 2a-h). Methane concentrations were lowest in February, when the highest concentrations in the cored sediment were 1.5 mM at station H6 and only 0.1 mM at station B1. The measured methane concentrations never exceeded the solubility limit for methane calculated for the *in situ* pressure, which ranged from 9.6 to 11.9 mM during the different sampling periods. Generally, methane concentrations at station H6 increased linearly from the surface down to 10 cm depth. Below this depth they only increased slightly or remained constant. An exception to this trend was observed in February at station B1, when methane showed a concave-upward trend indicating active consumption of methane in the topmost 10 cm of sediment.

Sulfate concentration gradients changed considerably between the different seasons at both stations reflecting substantial changes in sulfate reduction rates over the observation period. At both stations, the sulfate concentration gradients were steepest in October, intermediate in April and August, and lowest in February indicating highest and lowest sulfate reduction rates in October and February, respectively (Figure 3 a-h). At station H6, sulfate was always depleted in the cored sediment interval,





albeit at substantially greater depth in February. Depletion already occurred at 5 cm depth in April and
October and at 9 cm depth in August, and sulfate concentrations showed a typically concave downward
gradient. At station B1, sulfate was never consumed completely and concentrations remained above 1.5
mM at the bottom of the core. Generally, sulfate decreased steeply from the surface down to 10 cm
depth in August and October. Below this surface zone there was an interval with nearly constant
concentrations down to 20 cm depth, below which sulfate decreased again to a concentration to about
1.5 mM. Despite some variability in the sulfate concentration profiles and a lower gradient in the
topmost centimeters in April and February, the sulfate concentrations at the bottom of the core were
similar during all observation periods.

**3.3 $^{35}$S-sulfate reduction rates**
In agreement with the sulfate concentration gradients, $^{35}$S-sulfate reduction rates were higher at station
H6 than at station B1 (Figure 4 a-h). At station B1, SRR ranged from 0.2 nmol cm$^{-3}$ d$^{-1}$ to 63 nmol cm$^{-3}$
$^{3}$ d$^{-1}$, while at H6 SRR were as high as 411 nmol cm$^{-3}$ d$^{-1}$. Organoclastic sulfate reduction dominated the
interval down to 10 cm. Depth-integrated sulfate reduction rates over the core length varied from 9.2 to
11.7 mmol m$^{-2}$ d$^{-1}$ at station H6 and 0.5 to 2.4 mmol m$^{2}$ d$^{-1}$ at station B1.
Two distinct sulfate reduction rate peaks were found at station H6, one at the surface and a second peak
between 10 cm and 15 cm depth. The latter is in the sulfate-methane transition zone and indicates that
in this depth interval the rates of anaerobic methane oxidation coupled to sulfate reduction exceeded
organoclastic sulfate reduction rates. Depth-integrated rates of sulfate reduction in the sulfate-methane
transition zone at H6 were relatively constant and varied between 2.4 mmol m$^{-2}$ d$^{-1}$ and 2.8 mmol m$^{2}$ d$^{-1}$
(Table 2). In February, when sulfate penetrated to 24 cm depth, sulfate reduction rates were about two
times lower compared to the other months and a second sulfate reduction peak coupled to methane
oxidation was not visible. However, the distinct upward concave curvature of the methane profile in
February at station B1 indicates that even here some of sulfate reduction was coupled to anaerobic
methane oxidation and that this process overlapped with organoclastic sulfate reduction. Sulfate



reduction was also detected below the sulfate-methane transition zone at station H6 in April, August,
and October. Since non-radioactive carrier sulfate was added to the $^{35}$S-tracer during these incubations,
these rates indicate potential sulfate reduction activity in the methanogenic zone (Leloup et al., 2009).
The lack of the second peak in February at H6 is in agreement with previous observations that
productive seasons lead to shoaling of the methane-dependent sulfate reduction activity and anaerobic
oxidation methane layer in the sediments (Dale et al 2008, Treude et al 2005a). Previous studies at
neighboring stations H2 and H3 found AOM present at the depths 6-16 cm and 16-28 respectively,
which is in agreement with our findings (Wegener et al 2012).

**3.4 Benthic exchange of oxygen, sulfate, and methane**

Rates of total oxygen uptake are summarized in Table 2 and shown for comparison in Figure 5. Total
oxygen uptake was lowest in February at both stations (B1: -12±2.5 mmol m$^{-2}$ d$^{-1}$ and H6: -14.9±3.5
mmol m$^{-2}$ d$^{-1}$) and highest in August at station H6 (-26.9±3.7 mmol m$^{-2}$ d$^{-1}$) and in April at station B1 (-
33.5±4.7 mmol m$^{-2}$ d$^{-1}$). The diffusive sulfate fluxes from the water column into the sediment ranged
from -0.2 mmol m$^{-2}$ d$^{-1}$ in February to -1.4 mmol m$^{-2}$ d$^{-1}$ in October at B1 and from -1.3 mmol m$^{-2}$ d$^{-1}$ in
February to -2.7 mmol m$^{-2}$ d$^{-1}$ in August at station H6 (Table 2). These rates are significantly lower
than the radiotracer rates and indicate that sulfate is reoxidized below the sediment surface by reaction
with reactive iron (Thang et al., 2013). Methane fluxes determined by whole-core incubation were
consistently higher than the fluxes determined from the concentration profiles of dissolved methane at
station H6, whereas the two methods gave similar results at Station B1 (Table 2). The seasonal
variability in fluxes at the two stations was similar for the two measuring methods (Table 2). Whole-
core methane fluxes ranged from 0.3 mmol m$^{-2}$ d$^{-1}$ (February) to 19.9 mmol m$^{-2}$ d$^{-1}$ (August) at station
H6, and from 0.1 (February and April) to 1.2 mmol m$^{-2}$ d$^{-1}$ (August) at station B1 (Figure 5, Table 2).
The very high value measured in August 2012 at Station H6 is likely due to ebullition during the
incubation at ambient air pressure. Diffusive methane fluxes ranged from 0.05 mmol m$^{-2}$ d$^{-1}$ to 1.6
mmol m$^{-2}$ d$^{-1}$ at Station B1 and from 0.4 to 2.6 mmol m$^{-2}$ d$^{-1}$ in August at H6. The good agreement




between whole-core fluxes and diffusion-based fluxes at station B1 suggests that bioturbation and
irrigation at this station had little influence on the methane exchange with the bottom water.

**4 Discussion**
**4.1 Bottom water temperature and salinity**
Correlations between biogeochemical rates and fluxes with bottom water temperatures in
Himmerfjärden between April 2012 and February 2013 were weak for the period April-October, and
forced by the low rates in the coldest observation period in early February 2013. The temperature
versus rate/flux relationships were generally non-linear and not consistent for the fluxes of oxygen,
methane, and sulfate indicating that additional controlling factors played a role. It is likely that the
microbial community involved in the cycling of methane and sulfur species in Himmerfjärden sediment
is temperature-sensitive, and that the low rates in February 2013 are due to the 3°C temperature drop in
bottom water from October 2012 to February 2013. This would be consistent with rate observations in
comparable environments by Treude et al (2005a), Abril and Iversen (2002), Crill and Martens (1983),
and Westrich and Berner (1988), and is also supported by studies of the microbial community
composition of estuarine sediments that showed variations as a function of temperature (e.g., Zhang et
al 2014). However, microbial community composition and biogeochemical rates often cannot be
directly established from binary relationships with temperature, since other physical and chemical
parameters such as salinity, bottom water oxygen concentrations, organic carbon accumulation also
vary seasonally. Of these, salinity is not considered to be important for the present study, because the
annual range in Himmerfjärden bottom water was only between 5.4 and 7 ‰, which is too small to
affect the major electron acceptor and carbon degradation pathways.



## 4.2 Effects of organic matter composition and sedimentation


Organic carbon concentrations in Himmerfjärden are comparable to other fjord- and fjärd-type
estuarine sediments (Bianchi, 2007; Smith et al., 2015). Primary organic carbon export in
Himmerfjärden varies strongly on both seasonal and interannual timescales. The major export periods
occur during the spring phytoplankton bloom in March-April to early May, a late-summer
cyanobacterial bloom in August, and a secondary phytoplankton bloom in September (Bianchi et al.,
2002; Zakrisson et al., 2014; Harvey et al., 2015). Terrestrial-derived organic carbon that is not derived
from the sewage treatment plant plays only a minor role in this system, because no major rivers enter
the system and surface rainwater runoff is low. Based on sediment trap studies, the annual organic
carbon flux in Himmerfjärden varies by more than an order of magnitude at station B1 and by about a
factor of 3 in the inner parts of Himmerfjärden (Blomqvist and Larsson, 1994). However, only 10% to
60% of the total vertical mass flux may be composed of primary organic carbon, while the remainder
has been interpreted as resuspended material (Blomqvist and Larsson, 1994).
A second effect to be considered is that stations B1 and H6 are located in bathymetric depressions. H6
is in the center of a sub-basin separated from the outer Himmerfjärd by a sill (Fig. 1). Likewise, Station
B1 is located in a small depression at the head of a submarine channel that opens to the Baltic Sea.
Fine-grained and reworked organic-rich material preferentially accumulates in these depressions
(Jonsson et al., 2003). Because of the importance of resuspended organic material for the vertical mass
flux and bioturbation, the annual variability in the organic matter composition at the sediment surface
varies year-round only between 5 and 6 % OC with relatively constant C/N ratios between 7.9 and 9.1
at Station B1 and 8.3 and 9.2 at Station H6 (Bonaglia et al., 2014). The combined effect of the
sedimentation characteristics is that temporal variability in the bottom settling primary organic carbon
flux is low, which reduces the overall temporal variability in organic carbon amount and composition
and thereby in carbon mineralization rates. This small temporal variability is further influenced by
macrofauna bioturbation in the top 5 cm of sediment in this area, foremost by the bivalve *Macoma*
*baltica*, the arthropod *Pontoporeia femorata*, and the polychaete *Marenzelleria* (Bonaglia et al., 2014).



Although macrofauna is largely absent at Station H6, sediment is also mixed at station H6 by
bioturbating meiofauna (mostly ostracods) (Bonaglia et al., 2014).
The measured benthic oxygen uptake rates are consistent with the low variability in the surface organic
carbon concentrations, C/N ratios, and a temperature-dependent decrease in total oxygen uptake rates in
winter. The slightly higher total oxygen uptake rate at Station H6 is also consistent with the
physiography of the enclosed small basin favouring sediment trapping of fine material. In addition, the
location of station H6 in the inner fjärd limits water exchange and leads to greater oxygen depletion,
whereas the more open station B1 is affected by upwelling of oxygen-rich waters and comparatively
less burial of organic material (Table 1).

**4.3 Methane fluxes, sulfate reduction and methane oxidation**
The inner Himmerfjärden sediments have very high sedimentation rates between 0.9 and 1.3 cm/yr
(Thang et al., 2013; Bianchi et al., 2002). In such sediments organic carbon burial and transfer of
organic matter into the methanogenic zone is efficient and will occur within 20 to 30 years. As a
consequence of the low salinity (< 6 ‰) of the Baltic Sea at this latitude, seawater sulfate
concentrations are less than 7 mM and, by comparison with normal seawater, a comparatively lesser
amount of organic matter can be degraded by bacterial sulfate reduction (Thang et al., 2013).
Consequently, compared to normal marine sediment a larger proportion of organic matter undergoes
anaerobic microbial degradation terminating in methanogenesis, which generates a high upward flux of
methane into the sulfate-containing zone. Organiclastic sulfate-reducing bacteria will compete for the
available sulfate with sulfate-reducing bacteria involved in the anaerobic oxidation of methane (Dale et
al., 2006; Jørgensen and Parkes, 2010). Thermodynamic and kinetic constraints decide on the outcome
between these two competing processes. Dale et al. (2006) suggested that due to lower winter
temperatures and greater sulfate availability in the sulfate-methane transition zone in winter, the
thermodynamic driving force for anaerobic methane oxidation increases allowing for a greater
proportion of anaerobic methane oxidation coupled to sulfate reduction in the winter. In the summer



and fall, higher temperatures and sulfate limitation favor organiclastic sulfate reduction and
methanogenesis while limiting the anaerobic oxidation of methane. Most importantly, however, their
analysis showed that due to thermodynamic constraints and slow growth rates of the methane-oxidizing
archaea the microbial biomass does not change significantly over a year. These conceptual modelling
results can be tested with our Himmerfjärden data.
Sulfate reduction rates, particularly at H6, demonstrate how strongly bottom-water oxygen controls
organic matter mineralization. In the spring, summer, and fall sulfate reduction was at its maximum in
the first two centimeters of the sediments (Fig 3 e, f, g). In February, reduced organic carbon input and
higher oxygen concentrations resulted in lower sulfate reduction rates and a shift of the maximum rates
to greater depths in the sediments (Figure 3 h). Since other terminal carbon-oxidizing processes (e.g.
denitrification, iron, and manganese reduction) outcompete sulfate reduction for electron-donating
substrates, the depth of sulfate penetration and organic matter degradation via sulfate shifts deeper in
the sediment which reduces methane production.
The decrease in oxygen uptake matches well with the decrease in methane fluxes at the two stations in
winter, which suggests an impact of oxygen on methane cycling (Table 2, Figure 5). Higher oxygen
levels enhance bioturbation and oxygen uptake by the abundant macro- and meiofauna (Norkko et al.,
2015), but the mixing of sediment also affects methane transport to the water column, as the main
transport process shifts from diffusion to advection. This effect is likely the main cause for the winter
decrease in methane fluxes and concentrations. More aerated conditions indirectly enhance methane
removal by sustaining aerobic methanotrophs (Valentine 2011). It is plausible that, as in other brackish
coastal sediments, aerobic methanotrophs at the surface of Himmerfjärden sediments consume a
significant part of upward-diffusing methane that was not oxidized by anaerobic methane oxidation
(McDonald et al 2005, Moussard et al 2009, Treude et al 2005a).
Published benthic methane fluxes for estuaries with similar salinities have a reported range of 0.002 to
0.25 mmol m$^{-2}$ d$^{-1}$ (Abril and Iversen, 2002; Martens and Klump, 1980; Sansone et al., 1998; Zhang et
al., 2008; Borges and April, 2012; Martens et al., 1998). The methane fluxes derived from our core



incubations (0.1-2.6 mmol m$^{-2}$ d$^{-1}$, ignoring the potentially biased value of 19.9 mmol m$^{-2}$ d$^{-1}$) were
high compared to these published fluxes. Our fluxes are consistent with fluxes based on porewater
gradients by Thang et al. (2013) that were between 0.3 and 1.1 mmol m$^{-2}$ d$^{-1}$ at 3 nearby stations
measured in May 2009.
A conspicuous property of all porewater profiles at station H6, with the exception of the February 2013
sampling period, was the absence of a concave upward curvature in the methane concentration profiles,
which would be expected for net methane oxidation by aerobic and anaerobic methane oxidation
(Martens et al., 1998). Most concentration profiles of sulfate and methane at Station H6 overlapped
without a significant change in the methane concentration gradient. A similar observation has been
made earlier for other Himmerfjärden sediments (Thang et al., 2013), and has also been reported for
sediments of the northwestern Black Sea shelf (Knab et al., 2009) and in organic-rich shelf sediment of
the Namibian upwelling system (Brüchert et al., 2009). Inefficient methane oxidation is also evident
from the diffusive fluxes, which showed that the upward fluxes of methane into the sulfate-methane
transition zone were only marginally higher than the methane fluxes to the sediment surface indicating
little attenuation of the methane flux in the sulfate-methane transition zone (Table 2). One possible
explanation for this phenomenon is therefore that rates of sulfate reduction–coupled anaerobic methane
oxidation, except for the winter months, were low compared to the total sulfate reduction rate. An
alternative explanation of our observations could be that the methane concentration gradients were
affected by the presence of rising methane bubbles (Haeckel et al., 2007), or that bioturbation and
bioirrigation linearized the concentration profiles (Dale et al., 2013). However, we do not favor this
interpretation because of the absence of large macrofauna at station H6 and the fast porewater methane
sampling method.
An analysis of the cumulative distribution of $^{35}$S-SRR with depth at station H6 provides clues to the
proportion of organoclastic relative to anaerobic methane oxidation-coupled sulfate reduction at Station
H6 (Figure 6 e-h). The gradient in organoclastic sulfate reduction is well described by an exponential
law



$^{35}SRR = y\, z^{-b}$ (6)
where z is depth (cm) and y and b are regression coefficients (Jørgensen and Parkes, 2010). For the
sediments investigated here, the exponential coefficient b varied between 0.4 and 0.9 at station B1 and
0.3 and 0.8 at Station H6 (Table 4). At Station H6 the lowest coefficient was found for February 2013,
when sulfate penetrated the deepest into the sediment. Since the upward flux of methane provides an
additional energy source to sulfate-reducing bacteria, sulfate reduction rates are expected to increase in
the sulfate-methane transition zone. The net effect of a substantial AOM contribution to total sulfate
reduction is a low exponential coefficient b because the depth gradient in the sulfate reduction rate is
reduced. The difference between the exponential coefficients of the different sampling periods can be
used to calculate the variation in the contribution of AOM to the total sulfate reduction rate. At station
H6, between 5 % (August 2012) and 20% (April 2012) of the total sulfate reduction can be associated
with anaerobic methane oxidation. A comparison of the above method with the integrated $^{35}$S-sulfate
reduction rates integrated over the H6 sediment cores with the rates integrated over the AOM zone also
indicated that >20% of sulfate respiration at H6 was fuelled by methane (Table 2). In near-shore
continental margin sediments worldwide, the fraction of methane-driven sulfate reduction varies
between locations and accounts for 3-40% of total SRR, with 10% possibly representing a global mean
value (Jørgensen and Kasten, 2006). The average 20% contribution calculated here falls in the upper
range of these values and is similar to values reported before for one of the monitoring stations within
Himmerfjärden (Thang et al., 2013) and also for a very productive Chilean slope sediment (8-24 %)
(Treude et al 2005b). The good match between the upward fluxes of methane in the sulfate-methane
transition zone and the measured sulfate reduction rates in the transition also indicate that other
proposed electron acceptors for anaerobic methane oxidation such as iron are unimportant in these
sediments (Beal et al., 2009; Egger et al. 2014).





### 4.4 Temporal variability in hydrostatic pressure

The abrupt decrease in porewater methane concentrations from November 2012 to late January/early

February 2013 and the subsequent increase in April 2013 cannot be explained by variation in methane

oxidation alone, because the temporal change in porewater methane concentration was large compared

to the inferred methane oxidation rates based on fluxes in and out of the AOM zone. In addition, except

for downward-diffusing sulfate, there was no significant other electron acceptor present at depth. It is

unlikely that rates of methanogenesis would have decreased significantly between the fall and the

winter and resumed again in the spring, because the sediment temperatures were similar in February

and April (Table 1). Changes in organic matter sedimentation at the sediment surface also have no

significant influence on methanogenesis in buried sediment and cannot explain the sudden decrease in

methane concentration at depth. An alternative explanation for the changes in methane concentrations

is required. A possible explanation could be that changes in upward transport of methane changes are

due to variability in hydrostatic pressure and the associated diffusive and advective upward transport of

methane from depth. The free gas depth of methane is thought to follow changes in hydrostatic pressure

and temperature (Mogollon et al., 2011; Toth et al., 2015). An estimated 10% of the fine-grained

sediments in the Stockholm archipelago area is underlain by pockets of free methane (Persson and

Jonsson, 2000) and these free gas pockets are preferentially located in areas with the thickest

postglacial mud accumulation, generally in the center of the sub-basins and along fault lineaments

(Söderberg and Floden, 1992). Based on sub-bottom echosounder profiling, the surface of the free gas

zone in accumulation areas in Himmerfjärden is between 1 and 3 meter depth. During low sealevel

stand the free gas zone is expected to migrate closer to the sediment surface, whereas during high

sealevel the free gas zone is depressed into the sediment. The total variation in sealevel may as much as

50 cm, sufficient to explain the changes in methane concentrations observed here. Unfortunately, data

on sealevel fluctuation is not available for our respective sampling locations, and general sealevel

stands should not be directly applied to the sample sites.





**5 Conclusions**

A greater understanding of methane emissions from estuarine and coastal sediments is important to estimate the contribution of these environments to global marine methane fluxes. High benthic fluxes of methane from these sediments showed that aerobic and anaerobic methane oxidation rates are relatively inefficient, while still contributing up to 20% to total sulfate reduction. Higher bottom water oxygen concentrations in winter played a pivotal role in methane removal in these sediments. Of the different environmental regulators, bottom water oxygen had the strongest influence for the regulation of methane emissions. Oxygen availability directly enhanced aerobic organic matter mineralization by shifting the redox cascade in the sediments and indirectly by stimulating meiofauna and macrofauna activity thereby stimulating both the aerobic carbon mineralization and oxidative recycling of sulfate. The annual variability in sediment methane concentrations and benthic methane fluxes indicate that the annual environmental changes at these near-shore, but relatively deep-water localities are considerable. Very few data on sediment biogeochemical processes are currently available for aerobic and anaerobic carbon mineralization and methane cycling during winter months when ice cover inhibits access and sampling. Process rates inferred from sampling during open-water conditions over the whole year are therefore likely overestimates. In addition, advective recharge of subsurface methane should also be considered as an important transport component in deeper near-shore waters.



**6. Author contribution**
Joanna E. Sawicka conducted the sampling and analysis for the study and wrote the manuscript. Volker
Brüchert devised the study, interpreted the data, created the figures and tables, and wrote the
manuscript.

**7. Data availability**
The data are available from the second author upon request.

**8. Acknowledgments**
We are grateful to the staff of Askö Laboratory for their help and cooperation during the cruises and
our stays on the island of Askö. We would like to thank Barbara Deutsch, Camilla Olsson and Stefano
Bonaglia for their help during sampling. The study was funded by the grant from the Bolin Centre for
Climate Research, Baltic Ecosystem Adaptive management (BEAM), and the EU BONUS project
Baltic Gas.




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



**Table 1**. Main site characteristics of the sampling stations.

| Station | Sampling time | Water depth (m) | Temperature (°C) | Bottom water salinity (‰) | Bottom water Oxygen (µM) | Surface organic carbon (%) |
|---|---|---|---|---|---|---|
| | April 2012 | | 2.4 | 6.5 | 160 | 6.0 |
| **B1** 58°48'18''N 17°37'52''E | August 2012 | 41 | 6.9 | 7.0 | 260 | 5.2 |
| | October 2012 | | 6.8 | 7.0 | 224 | 5.1 |
| | February 2013 | | 3.4 | 7.0 | 380 | 5.0 |
| | April 2012 | | 1.8 | 5.9 | 40 | 4.6 |
| **H6** 59°04'08''N 17°40'63''E | August 2012 | 39.5 | 6.7 | 6.4 | 150 | 5.1 |
| | October 2012 | | 9.4 | 6.5 | 191 | 5.2 |
| | February 2013 | | 1.8 | 5.4 | 300 | 4.7 |








 

**Table 2.** Summary of $CH_4$ and $SO_4^{2-}$ fluxes, depth-integrated $^{35}SRR$, and total oxygen uptake (TOU).

| Station | Sampling time | Flux (mmol m$^{-2}$ d$^{-1}$) | | | | | | |
|---------|---------------|------|------|------|------|------|------|------|
| | | TOU | CH$_4$ | CH$_4$ | CH$_4$ | SO$_4^{2-}$ | $^{35}$S-SRR integrated over AOM[3] zone (n=3) | Integrated $^{35}$S-SRR (n=3) |
| | | whole core incubation (n=4) | whole core incubation (n=4) | Diffusive flux out of sediment (n=1) | Diffusive flux into SMTZ (n=1)[2] | Diffusive flux into sediment (n=1) | | |
| **B1** | April 2012 | -19.7 | 1.2 | 1.6 | | -0.4 | no AOM zone[4] | -2.3 |
| | August 2012 | -22.5 | 1.2 | no data | | -0.8 | no AOM zone[4] | -0.5 |
| | October 2012 | -21.1 | 1.9 | 1.9 | | -1.3 | no AOM zone[4] | -2.0 |
| | January 2013 | -12.0 | 0.1 | 0.1 | | -0.2 | no AOM zone[4] | -2.2 |
| **H6** | April 2012/13 | -23.5 | 3.9[1] | 2.2 | 2.8 | -2.6 | (10-15 cm)= 2.8 | -11.6 |
| | August 2012 | -26.9 | 19.9[5] | 2.4 | 2.6 | -2.5 | (10-15 cm) = 2.8 | -11.7 |
| | October 2012 | -25.9 | 1.8 | 1.8 | 1.9 | -2.6 | (10-15 cm)=2.4 | -11.5 |
| | January 2013 | -14.9 | 1.7 | 0.1 | 0.4 | -1.3 | no AOM zone[3] | -9.2 |

[1] whole core incubation was performed in April 2013; Diffusive fluxes were calculated for samples collected in April 2012;
[2] SMTZ - sulfate methane transition zone, [3] AOM zone – zone of anaerobic oxidation of methane, [4] no AOM zone means
that AOM zone was probably deeper than the core length; [5] potentially elevated due to depressurization/ex-solution
effect during core incubation at atmospheric pressure;



**Table 3.** Best-fit regression coefficients a and b for the depth gradient of sulfate
reduction rates ($^{35}$SRR = az$^{-b}$ (z =depth, cm)).

| Station | Sampling time | Exponential coefficient (a) | Exponential coefficient (b) |
|---|---|---|---|
| **B1** | April 2012 | 147.0 | -1.4 |
| | August 2012 | 11.7 | -0.9 |
| | October 2012 | 16.0 | -0.4 |
| | February 2013 | 33.5 | -0.8 |
| **H6** | April 2012 | 18.6 | -0.5 |
| | August 2012 | 37.4 | -0.5 |
| | October 2012 | 133.2 | -0.8 |
| | February 2013 | 25.0 | -0.4 |







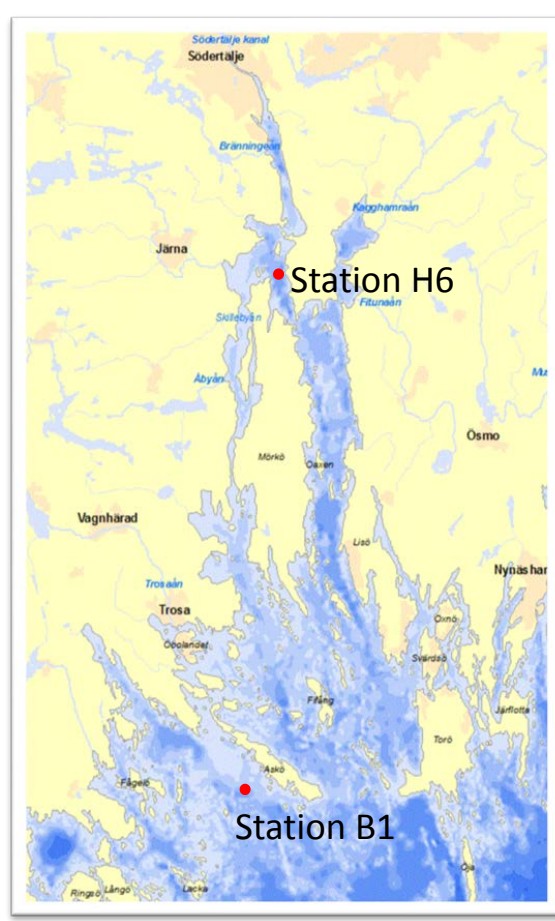

Figure 1. Location of sampling sites in
Himmerfjärden, Stockholm Archipelago, Sweden.
Detailed studies were conducted at two sites, an
open water site (Station B1) and in the inner part
of the estuary (Station H6).



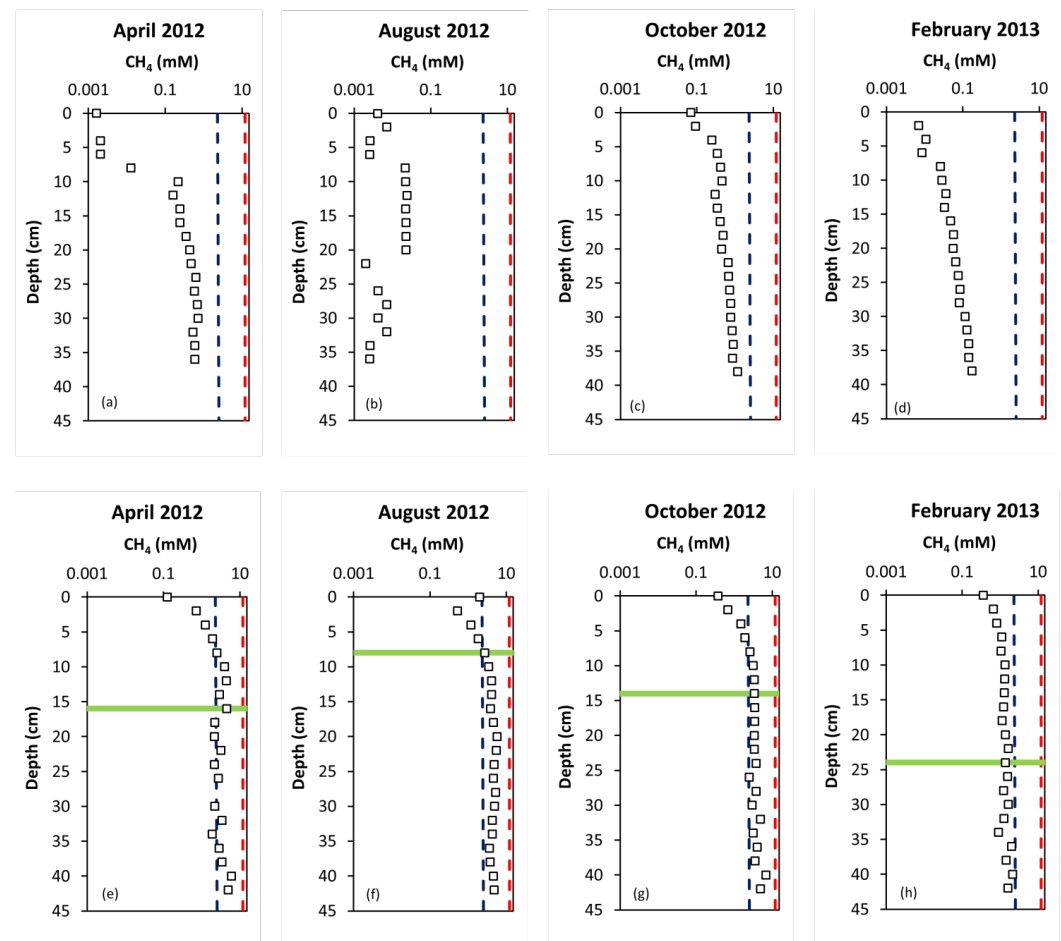



Figure 2. Porewater profiles of total methane at Station B1 (a-d) and Station H6 (e-h) for the different sampling periods. The green line marks the maximum depth of sulfate penetration. The dashed lines indicate the methane saturation concentration at 1 atm pressure (grey) and at the seafloor hydrostatic pressure (red) at the time of sampling.




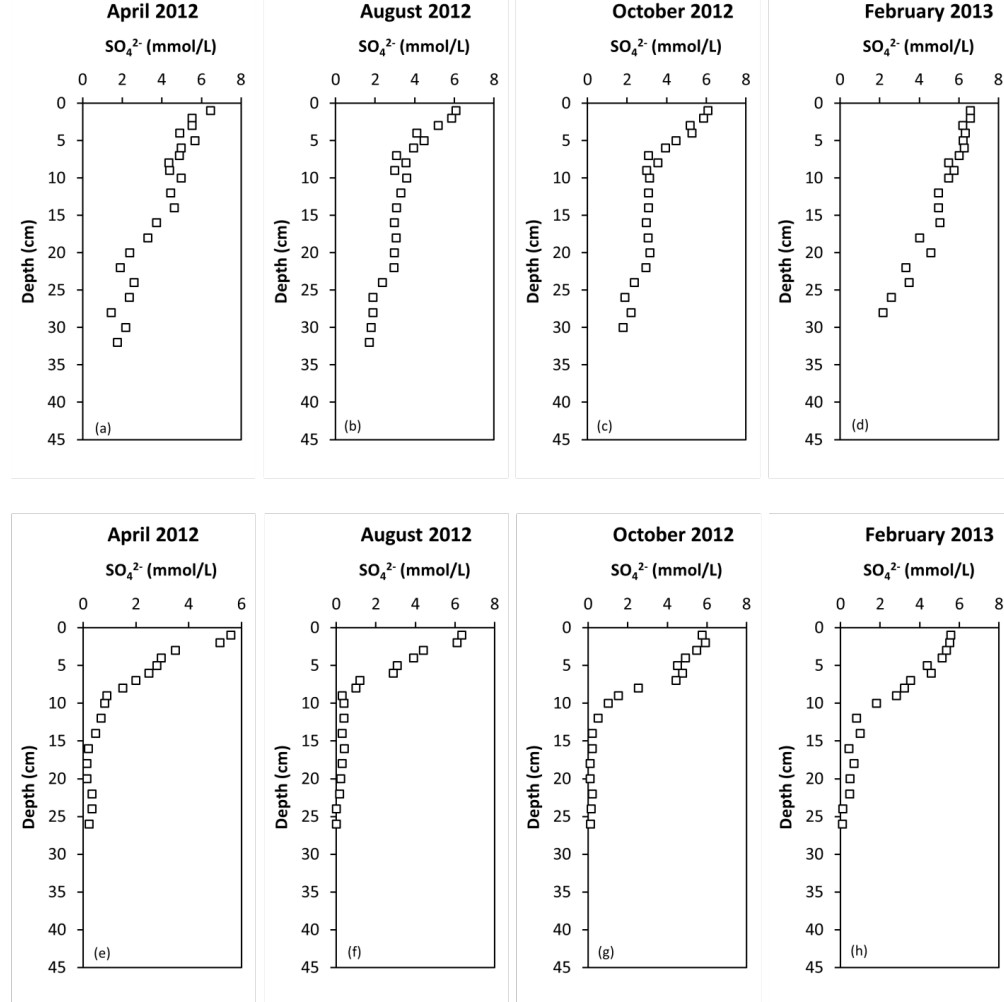


Figure 3. Porewater profiles of dissolved sulfate at Station B1 (a-d) and Station H6 (e-h) for the different sampling periods.



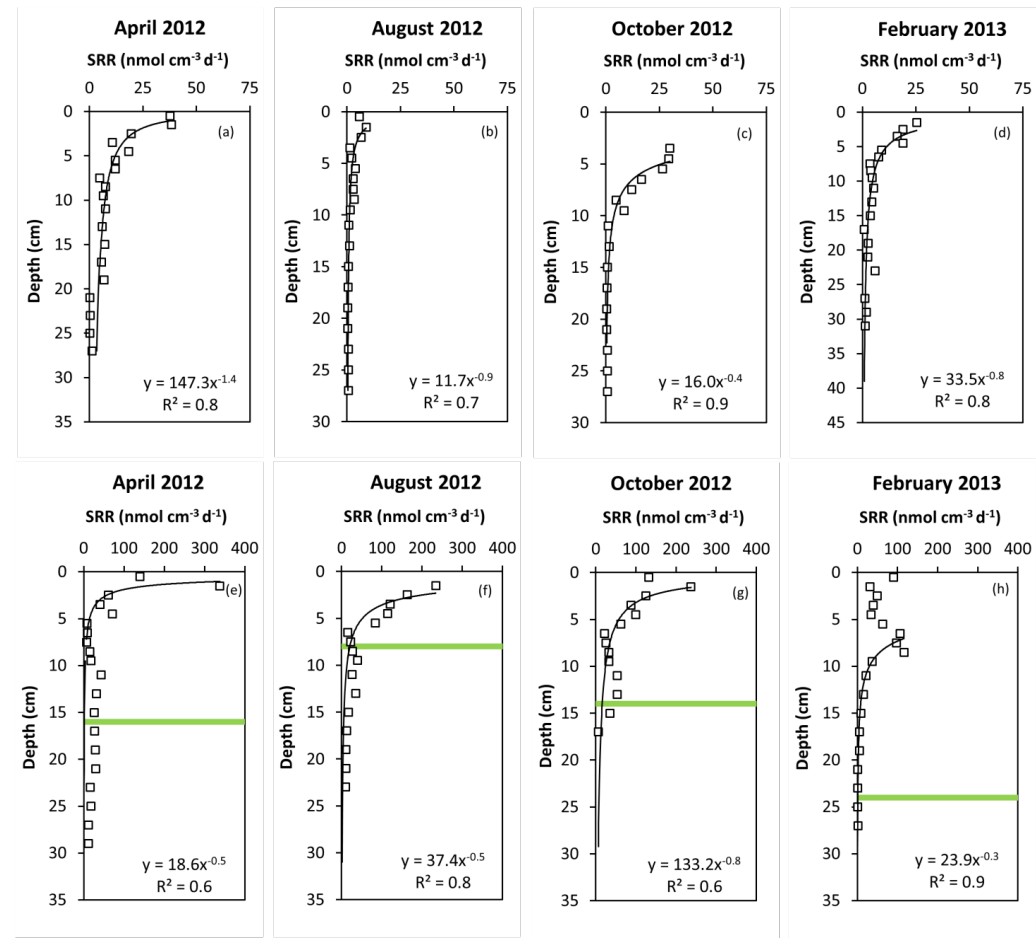

Figure 4. Depth gradients of bacterial sulfate reduction rates (SRR) measured with [35]S-sulfate. Black lines show the regression results to a power law of the form $y = ax^{-b}$. The green line marks the maximum depth of sulfate penetration.




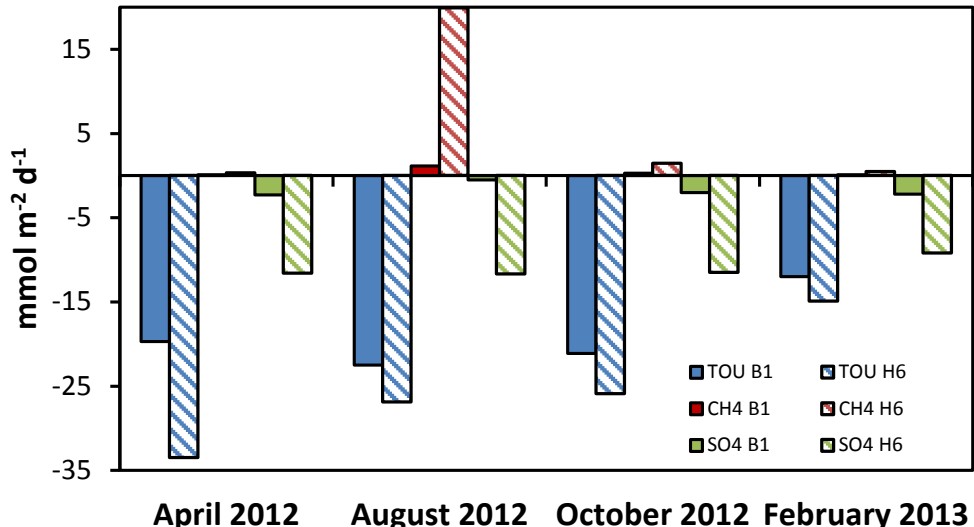


Figure 5. Comparison of benthic fluxes (mmol m$^{-2}$ d$^{-1}$) for sulfate (SO$_4$), methane (CH$_4$), and oxygen for the different sampling periods.





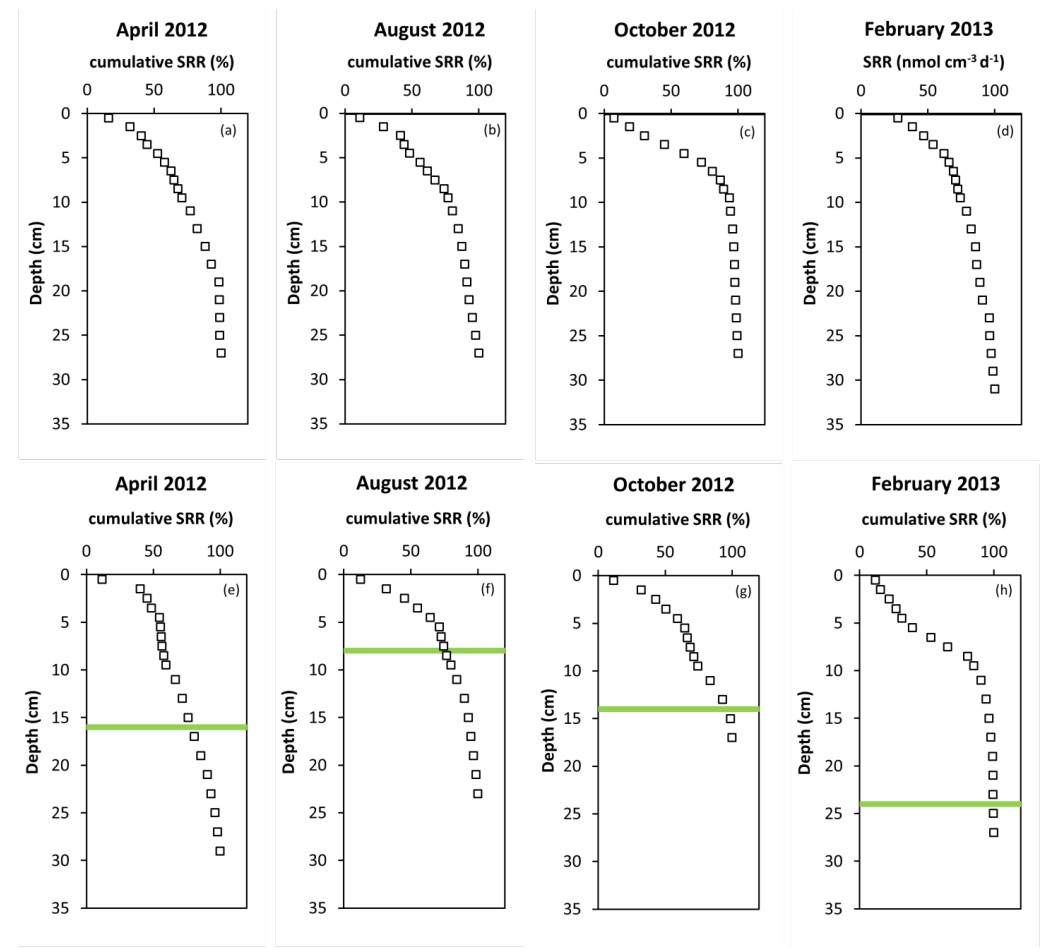


Figure 6. Depth distribution of sulfate reduction rate expressed as cumulative percentage. The green line marks the maximum depth of sulfate penetration.