# Peer review of "Seasonal variability and regulation of methane and sulfate"

_Biogeosciences, 2016_

## Referee Comment (RC1) · Anonymous Referee #1 · 19 Aug 2016

1.) General Comments

The paper investigates the effects of changes in temperature, benthic oxygen concentration and eutrophication on the sediment concentrations and fluxes of methane (and sulfate) in two sites in the Baltic Sea, an open-water coastal site and a eutrophic estuarine site over 4 time points (spring, summer, fall and winter) within a 12-month season. In order to address this, the authors measured methane and sulfate concentrations, oxygen uptake and sulfate reduction rates and calculated sulfate and methane fluxes in the sediment in the upper approx. 45 cm of the sediments. The main influence on methane emission from the sediment was found to be by bottom water oxygen enhancing aerobic carbon mineralization and oxidative recycling of sulfate. The authors

state that the seasonal changes in sediment methane concentrations are too large to be only the result of changes in methane generation and oxidation. Thus, they suggest advective recharge of methane from deeper, gas-rich sediment layers as possible influencing factor. The methane concentration below the sediment surface is lowered by AOM below the saturation concentration and thus bubble emission does not play a role at the investigated sites.

The study presents a well-designed experimental set-up and the experiments are performed thoroughly. However, the authors fail to formulate a clear scientific objective to conduct this research. It should be clear from the abstract and from the introduction why this study was conducted and what the expected merit would be. The abstract describes the findings and ends with the conclusion but it does not clearly mention the scientific questions addressed. At the beginning, the importance of this study should be made clear to attract the reader attention and interest, e.g. by naming the research question behind. Such questions can then be answered by the findings.

The introduction is well written, describes the state of the art and highlights some gaps in knowledge to justify the study. It also briefly summarizes the methods applied in the study. However, a concise statement what the presented study will contribute would make the paper sound much stronger. As mentioned above, it would be good if the authors state what problem they exactly address and how they do it – in other words, what exactly do they want to find out by the applied methods

The presentation of the results is confusing. There is major work needed to check the consistency of the figures and the text (see specific comments below). This makes it hard to follow the argumentation as one cannot relate the described results to the profiles. When presenting the results, I would suggest sticking to the same order of the stations throughout the entire manuscript. For example, always describe station B1 first and then station H6 and have the same order also in the tables and figures (i.e B1 on top and H6 below). The whole section should be rewritten with a focus to guide the reader clearly through the graphs. More attention should be paid to the general

consistency in the style of units. For example: mM vs. mmol/L vs. mmol L-1 (or e.g. nmol cc-3 d-1 and mmol L-1 OR nmol/cm3/d and mmol/L).

I suggest combining Figure 2 and 3 by plotting CH4 and sulfate concentration in the same plot with linear concentration scale also for methane. The logarithmic scale for methane makes it hard to follow the changes and it is easily to compare with the sulfate profile if both are on the same scale and together. I also suggest showing all data of the triplicates for the sulfate reduction rates in Figure 4 and making the fit - not only from the medians (see details below).

The interpretation and argumentation as well as the conclusions seem reasonable and are well written. The conclusion contains many good arguments and statements of which I think it would be good to mention these in the abstract to raise the interest of the reader. I suggest publication of this interesting study in Biogeosciences, however, I indicated major revision because the results presentation needs some careful rewriting with better guidance for the reader as well as careful cross-checking of text and figure/table content.

I think it is worth to add than 3 key words, to help finding the paper.

2.) Specific Comments

Lines 30-32: rephrase the sentence, and maybe split. At the moment it says that "The effects of temperature [. . .] where investigated [. . .] for open-water coastal and [. . .] sediment." That is probably not what the authors wanted to say.

Line 68: I would delete "summer"

Figure 1: is of rather bad quality (at least in the document I could print out). It is impossible to read the names of rivers, cities or islands. Maybe the colors could also have more contrast to make the whole picture look sharper. A color code/legend could be helpful to understand the different blue tones (is this water depth?). If this differentiation is not important, a single color for water would be better.

Lines 127, 129, 133, 148/149 : name equipment manufacture here Lines 180 – 185: total reduced inorganic sulfur should be abbreviated with TRIS, at least it should be consistent with the formula.

Line 189: I think it is better not to use the median here. Out of three measurements (triplicates) the median will always be the measurement in the middle. This means your plot and the input for the mathematical fit only relays on the one measured result (although there is the information behind that there is one higher and one lower measurement). You might talk about mean values in the text but in general I think you should present all individual measurements and also plot all data in the plots in Figure 5. And then you can calculate a fitted cure and also include this in the plot to visualize trends. It might be that the individual measurements show outliers and individual replicates differ. However, this is not uncommon for rate measurements and the best strategy is to simply show all data. Otherwise it could make the impression that something was tried to hide behind the median.

Line 245/256: be consistent in the order of described results (e.g. B1 as fist and H6 as second) within the text for all parameters and also with table 1.

Line 262ff: Please indicate the individual figure numbers after each station and result, e.g. "August... at station H6 (Fig. 2f) and . . .at station B1 (Fig. 2b) and so on for all mentioned data, this helps to identify the results in the figures. Please also make the order consistent overt the entire manuscript. Moreover, here are some inconsistencies between text and figure that could be easily sorted out by referencing to the respective profile. For example, in the text is says highest CH4 concentrations in August (H6: 5.7 mM, B1 1.9 mM). While for Station H6 (Fig. 2f) this might be true, for station B1 (Fig 2b) the figure I cannot see the 1.9 mM, in fact August 2012 has the lowest methane concentrations. Also in February, only at H6 (Fig, 2h) the CH4 concentration is lowest, but not at B1 (Fig. 2d). However, the number mentioned in the Text for B1 Feb 2103 (0.1 mM max.) matches the highest values in the B1 Feb 2013 figure (Fig. 2d) but it is not the lowest CH4 concentration in B1, this is in August 2012 (Fig. 2b).

**BGD**

I am wondering about the use of a logarithmic scale for the methane concentrations, this is unusual for the presentation of sediment methane concentrations. The mentioned linear increase in sediments at H6 is not visible due to the logarithmic concentration scale and also not the described "concave upwards trend" for B1. Also the mentioned differences in maximum concentrations are not visible due to that scale. Here, a linear concentration scale would be better to visualize the concentration changes. It would furthermore allow for a better judgement of the data quality and the efficiency of the sampling protocol (in terms of potential methane loss). A linear scale would also be helpful to compare the data with the sulfate data and the maximum sulfate penetration depth indicated by the green line. When using the linear scale it could be a good idea to combine Methane and Sulfate Profiles (Figures 2 and 3) in one plot for each sampling point.

Line 269: "concave upwards trend" what is meant by this? This is very unusual for a profile description. Do you mean increase followed by decrease? Here also a linear concentration scale would help to understand.

Line 272ff: I do not see that the sulfate concentration gradient at station H6 in October 2012 (Fig. 3g) . For me it seem that the steepest increase is in August (Fig.3f) (>6 mM over < 10 cm depth)

Line 275: better : "At station H6, sulfate was always fully depleted within the cores sediment interval, . . ."

Line 276: "Depletion already occurred at 5 cm depth in April and October and at 9 cm in August. . ." Depletion occurs all the way down from the surface sediment to the lowest concentration in the profile. Do you mean complete depletion (or depletion until a low constant level)? This is at approx. 9 cm depth at H6 in August 2012 (Fig. 3f) but I cannot see the 5 cm in April (Fig. 3e) and October (Fig. 3g), or do you refer here to Station B1 (Fig. 3a-d)?

Figures: 2-3 It would be helpful to quickly identify the profiles mentioned in the text,

if station number are indicated, e.g. for each row. The letters (a), (b), etc., should be larger in order to better overview the figure and relate it to the text while reading. The style of the units should be consistent with the format used in the text (mmol/L vs. mM). As mentioned in the comments above, I suggest combining figures 2 and 3 and presenting the methane concentration with a linear scale.

Line 287 ff: also here, please indicate the related profile in the figure 4 always directly when mentioned in the text, to help the reader understanding the text quickly.

Line 288/289: I don't see an SRR increase to 63 nmol cm-3 d-1 in any of the profile of B1. The maximum SRR I see is in Fig. 4a at approx. 35 nmol cm-3 d-1. Also for station H6, I do not find a maximum of 411 nmol cm-3 d-1 in the figure. The maximum measured is around 350 nmol cm-3 d-1 in Fig. 4e. Are these the individual measurements (i.e. from one of the triplicates?) As mentioned above, I suggest showing the data of all triplicate samples. If you refer to theoretical values at the very surface calculated from the regression line, please indicate so.

Line 305: What is the "peak between 6 and 9 cm depth? Isn't that a second peak? Sulfate is already mostly depleted at 10 cm and CH4 seems to be at maximum concentration below 10 cm. Could this increase SRR her not indicate AOM? Again, an overlay of sulfate and methane concentrations profile with linear concentration scale (combined Figs 2 and 3) would help to judge this better.

Figure 4: please indicate what H6 and B1, in the caption and best also in the Figure itself (e.g. for each line of plots). As mentioned earlier, I would like to see all individual data here instead of the median.

Figure 5: A separation line between the four sampling times would be helpful for a better readability.! Maybe also indicate them with Letters and reference to the plots in the text when described. Add the error bars if the errors are mentioned in the text. The figure says February 2013 but the Table January 2013

Line 314/315 and Fig 5: In the fig 5, highest TOU at H6 is in April (∼33 mmol m2 d-1) and at B1 in August ∼22 nmol cm-2 d-1) or so, contradictory to the text.

Line 315: sulfate flux seems to be lowest at B1 in August not in February and highest in February or April, contradictory to the text.

Line 398/399 ". . .constraints decide on the result of this competition between these two processes."

3.) Technical Corrections

line 144: cut-off

line 147: replace "to force out" by "to push"

line 149: "CH4 standards at 100 ppm and . . ."

line 156: (cm3) instead of (cubic centimeter)

line 162: missing dot after et al.

line 167: missing word after "adjacent"

line 177: 1 cm intervals

line 196: 40 L incubation tank

line 218: mL ("L" consistent to previous use)

line 221: replace "to force out" by "to push"

line 223: "CH4 standards at. . ."

line 227/228: 0.003 L / 0.009 L

line 257: 300 $\mu$M

line 467/468: remove one "integrated" in the sentence.

---

## Referee Comment (RC2) · Anonymous Referee #2 · 23 Aug 2016

Review "Annual Variability and regulation of methane in sulphate fluxes in Baltic Sea estuarine sediments." by Joanna E. Sawicka and Volker Bruechert

Sawicka and Bruechert study the seasonality of methane flux and sulphate reduction in two coastal sites in Sweden. With estuaries being important players in the global methane cycle, it is important to gain more insight into the controlling factors of methane oxidation in these systems.

Major comments:

There are several assumptions in the manuscript that are not backed up by either data or references. E.g. line 47-48, line 333-337, line 348, line 480-483, line 490-492, line

498-501, line 519-520.

The authors go back and forth about stating if the methane transport is controlled by diffusion or advection. Sediment permeability would help to understand what role advection can play. They state that changes in the hydrostatic pressure drives the changes in methane profiles, but do not explain what drives the changes in hydrostatic pressure, nor if that is related to season or not.

A lot of equations are listed that are just taken from other publications. Those do not need to be listed again.

It helps reading the manuscript if you keep the order of things the same, best throughout the manuscript (e.g. first mention station B1 then H6) but definitely in the same or consecutive sentences (e.g. Line 185-186, line 287 to 291). The order changes frequently in the manuscript making it harder to follow the arguments.

Minor comments

Title "Annual variability . . ." if it is mostly the pressure it seems that the year plays not an important role here, so I think annual is not very good. If it is the seasons, I think seasonal is better. ". . . Baltic Sea estuarine. . ." but you say later that you investigated an estuarine and an open water station. Better say coastal?

Abstract Line 41: You list 5.7mM as max in line 263 Line 43-4: ". . .lowering. . . far below the saturation concentrations." Your methane concentrations are also below the saturation concentrations below the sulphate penetration and seem to be mostly constant. Thus, the anaerobic methane oxidation does not seem to be lowering it far below the saturation concentration.

Introduction Line 55: Would be good to put the Tg into perspective to the global flux to know the importance. Line 66 and later in the methods: If the methane flux shows high spatial heterogeneity, why do you only measure the flux in one core? Line 70-1 and 517-9: You also say you do not have data from the ice covered period (line 126) Line

91: If you measure over four seasons, I feel seasonal is better than annual.

Materials and Methods: Line 102: Do you have info about the CH4, POM and DOM of the effluent of the STP? Line 113: How thick is the rusty brown surface layer? Line 113-116: Do you have information about the grain size or permeability? That would really be needed to argue for or against advective transport. Line 125-6: What did it mean that there was ice coverage? Line 127:"until for the experiment" change Line 133: "1N HCl" please change to 1M Line Line 134-5: Drying for 2 hours seems short. Did you test if longer drying had an effect? Line 139: "seconds" seems a bit over stated, especially knowing that it takes already probably more than seconds before the core was on the ship. "less that a minute" still sounds very impressive and is more realistic. Line 144: "exactly" delete Line 145: "5M NaCl" that is not a standard treatment, did you test if it halted microbial activity? Why did you not use base? Line 146: If you only leave the sample for 1 hour you will not get all the gas adsorbed to clay minerals. Did you do later measurements to determine if the concentrations were constant? Line 149: What column did you use on your GC? Line 162: "10% HCl" is that 10% concentrated HCl in water or is that a 3.7 dilution of concentrated (37%) HCl? Better give M concentrations. Line 174: Why did you add cold sulphate to the tracer solution? That introduces sulphate into the sulphate free zone and does not do much in the not sulphate free zone. Line 185: "($SO_4^{2-}$)"does not appear in the formula Line 188-189: If you only show the median, what is the error? Plot it in the graph, or if it pretty much constant, state it here. Line 215: Why do you use a different fixing agent for the methane samples? Why do you not use base? Line 225-229: The list of variables and the equation do not fit to each other. Line 233-4: Volumetric units do not match. Line 237-42: Did you see any signs of burrows in your cores? Why do you only do one replicate if you know that it is spatially heterogeneous? You assume here that diffusion is the dominant process? Based on what? Later you discuss that bioturbation can affect the transport and shift it from diffusive to advective. Line 248: "Do" why was it recalculated?

Results Line 263: In the abstract you state that methane concentrations exceeded 6mM. Line 267-270: it is hard to see linearity or concave shapes in log plots. If these are important don't use log plots. Line 271: You state that the methane profiles can not be explained by the T and Corg changes. How about the sulphate profiles? Line 303: You would also get a rate in the sulphate free zone if you would have injected tracer only. If you use the sulphate concentrations from you profile to determine the sulphate reduction rate? It does not matter that the tracer is reduced. If there is no sulphate the rate is still 0. Line 305-7: Why is that happening? Line 313-7: All those rates should not be negative. A negative oxygen uptake of the sediment means diffusion of oxygen out of the sediment. Did you test if you could detect changes in sulphate concentrations in the whole core incubations? Line 319: how about reoxidation by oxygen not only iron? Why do you suddenly have reactive iron here, when you state in line 484 that there is no other electron acceptor available? Line 319: You have to keep in mind that the methane profile does not really give you a rate of methane diffusion out of the sediment. If there is oxygen in the surface sediment there is likely aerobic methane oxidation as you discuss yourself. Thus, this is an over estimation of the methane flux, and you do not know by how much. You need to have methane concentration data at the scale of the oxygen consumption to determine that. Line 328-9: As your diffusion based fluxes are overestimated (see comment above) there is no agreement, and thus bioturbation and irrigation, as well as advection as a result of your stirring probably effects the flux.

Discussion Line 334-7: Do you have data available to support that? Can you model that it does not fit? Line 342: How about rate studies in temperature gradient blocks e.g. by Sagemann or Arnosti? Line 348: How big could the effect be with this difference in salinity? Line 352: How much Corg comes from the sewage treatment plant? How similar or different is the fjord thus to others? Line 357-9: Where do the high sedimentation rates come from if there is only low river runoff? Line 385: Table 1 has no information about the burial of organic material. Do you have depth profiles supporting that? Line 391: Salinity of B1 is 7‰ Line 392-4: A little too often "compar*" Line 392-8: Lower sulphate concentrations mean that there is less sulphate available for organoclastic & methanotrophic sulphate reduction, just by simple numbers. Line 410&2: Figure 4 Line 412-3: Iron and manganese reducers do not always outcompete sulphate reducers, see work by Thamdrup and Vandieken. Line 414-5: You state that the main driver for the differences is the advective flow based on the hydrostatic pressure, but here you speculate about more sulphate reduction leads to less methanogenesis. Which process is now the important one? Line 417-20: How deep is the bioturbation in these sediments? Line 419-21: In advective systems with bioturbation fluxes should increase not decrease. Line 443-5: This is not an explanation, it is just stating that you believe the data in contrast to the scenario below. Line 454: "law" replace with "function" Line 461-3: Sentence not clear Line 465: What is the percentage if you compare the methane flux into the SMTZ with the accumulated SRR or the total methane flux with the SRR? Do the numbers fit what model says? Line 474-7: In line 445 you state that there is only little link. Line 480-3: Please provide some support for this, maybe with a model. Line 484: Did you determine the concentrations of other electron acceptors like Fe? In line 319 you state that it is available for sulphide reoxidation. Line 490-2: Do you have any data on changing pressures? What would drive these changes? What would the possible magnitude be? Line 500-1: Do you have any data or reference to support this magnitude? Line 500: "may as much as. . ." insert "be" Line 497-503: If it is the hydrostatic pressure it is actually not really a seasonal effect?

Conclusion Line 508-9: Sentence not clear. Line 515-6: How is it seasonal/annual if it is the hydrostatic pressure? Not clear. Line 517-9: You also state that you do not have data from the ice covered times! Line 519-20: Why is that if the temperature and Corg input do not play an important role? Line 520-1: Is it no advective or diffusive transport that controls the methane? You keep changing your argument.

Tables Table 2: "no AOM zone3" for H6 January 2013, change to "no AOM zone4" Table 3: "Exponential coefficient (a)" this one is not exponential.

Figures Make all the y axis for depth the same scale. Figure 1: The colors do not help reading the map too much, not very clear in black and white at all. Change continent

to just white or black to make ocean more clear and maybe reduce shades in the water. Figure 2: You use the maximum sulphate penetration from figure 3 here. Maybe change the order of the figure to keep the flow consistent. Figure 3: How do you define your maximum sulphate penetration if there is a sulphate peak in your graphs below it? Figure 5: Keep order consistent between listing in the graph and in the caption. If you keep all the values positive it is much easier to compare the values. Which method are the methane fluxes based on?

---

## Author Comment (AC1) · 14 Sep 2016

1.) General Comments

The paper investigates the effects of changes in temperature, benthic oxygen concentration and eutrophication on the sediment concentrations and fluxes of methane (and sulfate) in two sites in the Baltic Sea, an open-water coastal site and a eutrophic estuarine site over 4 time points (spring, summer, fall and winter) within a 12-month season. In order to address this, the authors measured methane and sulfate concentrations, oxygen uptake and sulfate reduction rates and calculated sulfate and methane fluxes in the sediment in the upper approx. 45 cm of the sediments. The main influence on methane emission from the sediment was found to be by bottom water oxygen enhancing aerobic carbon mineralization and oxidative recycling of sulfate. The authors state that the seasonal changes in sediment methane concentrations are too large to be only the result of changes in methane generation and oxidation. Thus, they suggest advective recharge of methane from deeper, gas-rich sediment layers as possible influencing factor. The methane concentration below the sediment surface is lowered by AOM below the saturation concentration and thus bubble emission does not play a role at the investigated sites. The study presents a well-designed experimental set-up and the experiments are performed thoroughly.

However, the authors fail to formulate a clear scientific objective to conduct this research. It should be clear from the abstract and from the introduction why this study was conducted and what the expected merit would be. The abstract describes the findings and ends with the conclusion but it does not clearly mention the scientific questions addressed. At the beginning, the importance of this study should be made clear to attract the reader attention and interest, e.g. by naming the research question behind. Such questions can then be answered by the findings.

The introduction is well written, describes the state of the art and highlights some gaps in knowledge to justify the study. It also briefly summarizes the methods applied in the study. However, a concise statement what the presented study will contribute would make the paper sound much stronger. As mentioned above, it would be good if the authors state what problem they exactly address and how they do it – in other words, what exactly do they want to find out by the applied methods.

**Answer: We will reformulate our objectives and scientific and try to emphasize the scope of the study somewhat clearer.**

The presentation of the results is confusing. There is major work needed to check the consistency of the figures and the text (see specific comments below). This makes it hard to follow the argumentation as one cannot relate the described results to the profiles. When presenting the results, I would suggest sticking to the same order of the stations throughout the entire manuscript. For example, always describe station B1 first and then station H6 and have the same order also in the tables and figures (i.e., B1 on top and H6 below). The whole section should be rewritten with a focus to guide the reader clearly through the graphs. More attention should be paid to the general consistency in the style of units. For example: mM vs. mmol/L vs. mmol L-1 (or e.g. nmol cc-3 d-1 and mmol L-1 OR nmol/cm3/d and mmol/L).

**Answer: We will follow all the suggestions by the reviewer to improve the clarity.**

I suggest combining Figure 2 and 3 by plotting CH4 and sulfate concentration in the same plot with linear concentration scale also for methane. The logarithmic scale for methane makes it hard to follow the changes and it is easily to compare with the sulfate profile if both are on the same scale and together. I also suggest showing all data of the triplicates for the sulfate reduction rates in Figure 4 and making the fit - not only from the medians (see details below).

**Answer: The same comments were made by reviewer 2. We have gone back and forth in our consideration of the best visual presentation to reflect our thinking. We take the reviewers as reflecting that of the majority of potential readers and have revised the scale to show linear plots. Naturally, there is some noise in the data. The intention of the logarithmic presentation was to show the extreme drop in concentration and the variability between the different sampling times that is most relevant. While we think that the key information was also conveyed in the logarithmic plot, we follow the recommendation.**

The interpretation and argumentation as well as the conclusions seem reasonable and are well written. The conclusion contains many good arguments and statements of which I think it would be good to mention these in the abstract to raise the interest of the reader.

I suggest publication of this interesting study in Biogeosciences, however, I indicated major revision because the results presentation needs some careful rewriting with better guidance for the reader as well as careful cross-checking of text and figure/table content.

I think it is worth to add than 3 key words, to help finding the paper.
**Methane cycling, coastal and estuarine sediment, seasonality**

2.) Specific Comments

Lines 30-32: rephrase the sentence, and maybe split. At the moment it says that "The effects of temperature [: : :] where **(A: where ?)** investigated [: : :] for open-water coastal and [: : :] sediment." That is probably not what the authors wanted to say.
**Answer: We will try to simplify this sentence, but we are not exactly sure what the reviewer meant by this comment.**

Line 68: I would delete "summer"
**Done**

Figure 1: is of rather bad quality (at least in the document I could print out). It is impossible to read the names of rivers, cities or islands. Maybe the colors could also have more contrast to make the whole picture look sharper. A color code/legend could be helpful to understand the different blue tones (is this water depth?). If this differentiation is not important, a single color for water would be better.
**Thanks for the suggestion. We have produced an alternative map in Ocean Data View, ,however, without the rastered high-resolution bathymetry. We would have liked to retain the bathymetric information, because it is important to understand the sediment deposition pattern, but cannot provide it at an overall satisfying resolution, because the bathymetric data are rastered for a large coastal area and cannot be presented as high-resolution cutouts for Himmerfjärden.**

Lines 127, 129, 133, 148/149 : name equipment manufacture here
**Okay.**
Lines 180 – 185: total reduced inorganic sulfur should be abbreviated with TRIS, at least it should be consistent with the formula.

**Okay**

Line 189: I think it is better not to use the median here. Out of three measurements (triplicates) the median will always be the measurement in the middle. This means your plot and the input for the mathematical fit only relays on the one measured result (although there is the information behind that there is one higher and one lower measurement). You might talk about mean values in the text but in general I think you should present all individual measurements and also plot all data in the plots in Figure 5. And then you can calculate a fitted curve and also include this in the plot to visualize trends. It might be that the individual measurements show outliers and individual replicates differ. However, this is not uncommon for rate measurements and the best strategy is to simply show all data. Otherwise it could make the impression that something was tried to hide behind the median.
**A: The format was chosen to present the trends in the best possible way. We have revamped the figures and now show all data points.**

Line 245/256: be consistent in the order of described results (e.g. B1 as first and H6 as second) within the text for all parameters and also with table 1.
**We will carefully check the text to be consistent in the sequence the stations are listed.**

Line 262ff: Please indicate the individual figure numbers after each station and result, e.g., "August... at station H6 (Fig. 2f) and : : :at station B1 (Fig. 2b) and so on for all mentioned data, this helps to identify the results in the figures. Please also make the order consistent over the entire manuscript. Moreover, here are some inconsistencies between text and figure that could be easily sorted out by referencing to the respective profile. For example, in the text is says highest $CH_4$ concentrations in August (H6: 5.7 mM, B1 1.9 mM). While for Station H6 (Fig. 2f) this might be true, for station B1 (Fig 2b) the figure I cannot see the 1.9 mM, in fact August 2012 has the lowest methane concentrations. Also in February, only at H6 (Fig, 2h) the $CH_4$ concentration is lowest, but not at B1 (Fig. 2d). However, the number mentioned in the Text for B1 Feb 2103 (0.1 mM max.) matches the highest values in the B1 Feb 2013 figure (Fig. 2d) but it is not the lowest $CH_4$ concentration in B1, this is in August 2012 (Fig. 2b).
**Thanks for this suggestion. We will adapt the text accordingly.**

I am wondering about the use of a logarithmic scale for the methane concentrations, this is unusual for the presentation of sediment methane concentrations. The mentioned linear increase in sediments at H6 is not visible due to the logarithmic concentration scale and also not the described "concave upwards trend" for B1. Also the mentioned differences in maximum concentrations are not visible due to that scale. Here, a linear concentration scale would be better to visualize the concentration changes. It would furthermore allow for a better judgement of the data quality and the efficiency of the sampling protocol (in terms of potential methane loss). A linear scale would also be helpful to compare the data with the sulfate data and the maximum sulfate penetration depth indicated by the green line. When using the linear scale it could be a good idea to combine Methane and Sulfate Profiles (Figures 2 and 3) in one plot for each sampling point.

**A: We have reconverted the figures back to a linear scale. We were well aware of the potential criticism the use of a logarithmic scale can draw. However, logarithmic scales have been used frequently, e.g., in the IODP literature. Here the main purpose was to convey the scale of the concentration change and to demonstrate that the sampling technique allows us to capture concentrations above the concentration at atmospheric pressure with consistency, but that the measured concentrations were consistently far below the saturation concentration at in situ pressure. The linear scale does not allow us to do this. However, we welcome the honest opinion of the reviewer, and therefore reverted to the conventional linear scale.**

Line 269: "concave upwards trend" what is meant by this? This is very unusual for a profile description. Do you mean increase followed by decrease? Here also a linear concentration scale would help to understand.

**A: Scale was changed back to linear to avoid more confusion.**

Line 272ff: I do not see that the sulfate concentration gradient at station H6 in October 2012 (Fig. 3g) . For me it seems that the steepest increase is in August (Fig.3f) (>6 mM over < 10 cm depth)

**A: The steepest gradient occurs between 6 and 10 cm depth in October whereas the gradient is less steep. This is likely due to the increased O$_2$ concentrations and the colder temperature in the fall leading to a downward propagation of a less steep sulfate gradient, however, not yet at a depth below 6 cm.**

Line 275: better : "At station H6, sulfate was always fully depleted within the cored sediment interval, : : :"
**A: okay**

Line 276: "Depletion already occurred at 5 cm depth in April and October and at 9 cm in August: : :" Depletion occurs all the way down from the surface sediment to the lowest concentration in the profile. Do you mean complete depletion (or depletion until a low constant level)? This is at approx. 9 cm depth at H6 in August 2012 (Fig. 3f) but I cannot see the 5 cm in April (Fig. 3e) and October (Fig. 3g), or do you refer here to Station B1 (Fig. 3a-d)?
**The reviewer is correct. We will change this.**

Figures: 2-3 It would be helpful to quickly identify the profiles mentioned in the text, if station number are indicated, e.g., for each row. The letters (a), (b), etc., should be larger in order to better overview the figure and relate it to the text while reading. The style of the units should be consistent with the format used in the text (mmol/L vs. mM). As mentioned in the comments above, I suggest combining figures 2 and 3 and presenting the methane concentration with a linear scale.
**We will increase the numbers and assign Station names to each row. We also plot sulfate and methane together in one plot.**

Line 287 ff: also here, please indicate the related profile in the figure 4 always directly when mentioned in the text, to help the reader understanding the text quickly.
**We will do this.**

Line 288/289: I don't see an SRR increase to 63 nmol cm-3 d-1 in any of the profile of B1. The maximum SRR I see is in Fig. 4a at approx. 35 nmol cm-3 d-1. Also for station H6, I do not find a maximum of 411 nmol cm-3 d-1 in the figure. The maximum measured is around 350 nmol cm-3 d-1 in Fig. 4e. Are these the individual measurements (i.e. from one of the triplicates?) As mentioned above, I suggest showing the data of all triplicate samples. If you refer to theoretical values at the very surface calculated from the regression line, please indicate so.

**We will now show all replicates and show the regression lines based on the power law model. The author is correct in that assuming that the reference in the text was made for the individual measurements and not the averages.**

Line 305: What is the "peak between 6 and 9 cm depth? Isn't that a second peak? Sulfate is already mostly depleted at 10 cm and CH4 seems to be at maximum concentration below 10 cm. Could this increase SRR her not indicate AOM? Again, an overlay of sulfate and methane concentrations profile with linear concentration scale (combined Figs 2 and 3) would help to judge this better.

**A: We had of course also debated the second peak. It is a consistent feature, since it is also nicely visible in the 3 replicate SRR. However, there is no change in the sulfate gradient or the methane gradient in this depth intervals that would be expected if the measured rates would be largely attributed to AOM. Therefore, we consider it less likely that the SRR at this depth indicate largely AOM and intepret this peak as due to organiclastic SR, but we will add a sentence addressing this issue.**

Figure 4: please indicate what H6 and B1, in the caption and best also in the Figure itself (e.g. for each line of plots). As mentioned earlier, I would like to see all individual data here instead of the median.

**Okay. We will do so.**

Figure 5: A separation line between the four sampling times would be helpful for a better readability.! Maybe also indicate them with Letters and reference to the plots in the text when described. Add the error bars if the errors are mentioned in the text. The figure says February 2013 but the Table January 2013 Line 314/315 and Fig 5: In the fig 5, highest TOU at H6 is in April (_33 mmol m2 d-1) and at B1 in August _22 nmol cm-2 d-1) or so, contradictory to the text.

**We have changed the figure accordingly and adjusted the text.**

Line 315: sulfate flux seems to be lowest at B1 in August not in February and highest in February or April, contradictory to the text.

**We assume that the reviewer refers to the difference in uptake calculated for the $SO_4$ gradient and the SRR for August 2012. We have clarified in the figure 5 that the sulfate flux data are the $^{35}S$ data.**

Line 398/399 ": : :constraints decide on the result of this competition between these two processes."

3.) Technical Corrections
line 144: cut-off
**corrected**
line 147: replace "to force out" by "to push"
**revised**
line 149: "CH4 standards at 100 ppm and : : :"
**inserted 'of'**
line 156: (cm3) instead of (cubic centimeter)
**changed**
line 162: missing dot after et al.
**. added**
line 167: missing word after "adjacent"
**inserted 'intervals'**
line 177: 1 cm intervals

line 196: 40 L incubation tank

line 218: mL ("L" consistent to previous use)
**changed**
line 221: replace "to force out" by "to push"
**changed**
line 223: "CH4 standards at: : :"
**'of'**
line 227/228: 0.003 L / 0.009 L
**inserted the space**
line 257: 300 _M
**µM**

line 467/468: remove one "integrated" in the sentence.

**Removed the first 'integrated'**

Review "Annual Variability and regulation of methane in sulphate fluxes in Baltic Sea estuarine sediments." by Joanna E. Sawicka and Volker Bruechert

Sawicka and Bruechert study the seasonality of methane flux and sulphate reduction in two coastal sites in Sweden. With estuaries being important players in the global methane cycle, it is important to gain more insight into the controlling factors of methane oxidation in these systems.

Major comments:

There are several assumptions in the manuscript that are not backed up by either data or references.

E.g.

Line 47-48: Importance of advective processes.

**See comments below**

Line 333-337: Temperature regulation inference.

**A: Temperature regulation would imply that methane oxidation is less temperature-sensitive than methanogenesis preventing methane oxidizers from keeping up with the enhanced methane flux during summer. This requires significantly different in the $Q_{10}$ of methanogens and methane oxidizers. Publications from lake environments and terrestrial environments, e.g., King et al. (1988), Wik et al. (2016) Nguyen et al (2011) suggest that aerobic methane-oxidizing bacteria may have higher Q10 than methanogens, but this argument remains unproven for marine habitats. In case of anaerobic methane oxidation, it is difficult to argue for a temperature adaptation disadvantage compared to methanogenesis, because of the tight coupling between sulfate reduction and methane oxidation and the phylogenetic proximity of ANME and ANME to known methanogens, in particular with regard to membrane lipid composition, which should be considered the strongest physological regulator.**

Line 348 salinity variation.

**A: Data shown in Table 1 indicate that the salinity for the different sampling periods varied little.**

Line 480-483 Variability in methane concentrations not due to variability in methane oxidation rates alone

**A: The emphasis here is on the word alone. The whole first paragraph of the discussion emphasizes the different regulatory processes that affect methane concentrations and two important ones are of course temperature and bottom water oxygen.**

Line 490-492: Changes in the upward transport rate of methane

**A: Again, methanogenesis rates can only increase due a temperature increase, since the availbility of organic carbon for methane production in buried sediment does not change. These effects were modelled by Dale et al. (2006) and are discussed here.**

Line 498-501. Migration of the methane saturation zone due hydrostatic pressure changes

**A: There is acoustic echosounder evidence for free gas presence in these sediments and the authors have personal communcation (Tom Floden) evidence that the depth of the free gas zone as seen on the acoustic echosounder changes substantially from year to year. The mechanisms that affect gas migration in these sediment are manifold. They can have to do with atmospheric pressure changes, wind direction**

**affecting water levels and these in turn affect the solubility of methane at a given temperature. An additional parameter is groundwater movement. There is also geophysical evidence from other areas in the outer Himmerfjärden area that suggest groundwater seepage. This implies a complex aquifer hydrology that was accessible with the coring methods used here, but that has indirect effects on methane solubility, advective transport, and effective methane flux. A mechanistic evaluation of all these processes is far beyond the scope of this paper and cannot be adequately addressed in addition to the data presented here.**

Line 519-520. The period of ice cover has low flux rates. Extrapolation of rates during open-water conditions for a whole year would therefore be overestimates.

**A: To our knowledge, this is one of the very few studies that reports sulfate reduction rates in a fully ice-covered estuary. These rates were very low compared tot he open-water season. Extrapolation of these rates for the ice-covered period will necessarily lower the annually integrated rates.**

The authors go back and forth about stating if the methane transport is controlled by diffusion or advection.
Sediment permeability would help to understand what role advection can play.
**A: While this could help, we would like to point out that even low-permeability sediment emit bubbles.**

They state that changes in the hydrostatic pressure drives the changes in methane profiles, but do not explain what drives the changes in hydrostatic pressure, nor if that is related to season or not.
**Changes in hydrostatic pressure in the Baltic Sea are influenced by air pressure, prevailing wind direction, and general sealevel stand due to the balancing effects of saltwater entry through the Danish straits and freshwater discharge in the northern Baltic and from the rivers flowing in from the south. Additional effects are caused by the local coastal topography. These multiple parameters result in complex subsurface hydrology and complexity in estuarine water level conditions that make it difficult to use general meteorological observations to predict sealevel variability. Hydrograpic data are therefore reported for the general area only.**

A lot of equations are listed that are just taken from other publications. Those do not need to be listed again.
**Fine – this can be changed, but there are still many readers that are unfamiliar with some of the methods. Without going into too much detail, these equations provide the basic framework.**

It helps reading the manuscript if you keep the order of things the same, best throughout the manuscript (e.g. first mention station B1 then H6) but definitely in the same or consecutive sentences (e.g. Line 185-186, line 287 to 291). The order changes frequently in the manuscript making it harder to follow the arguments.
**We will carefully check the manuscript to make sure the sequence is adhered to consistently.**

Minor comments
Title "Annual variability : : :" if it is mostly the pressure it seems that the year plays not an important role here, so I think annual is not very good. If it is the seasons, I think seasonal is better. ": : : Baltic Sea estuarine: : :" but you say later that you investigated an estuarine and an open water station. Better say coastal?
**Thanks for pointing this out. We will change this.**

Abstract Line 41: You list 5.7mM as max in line 263 Line 43-4: ": : :lowering: : : far

below the saturation concentrations." Your methane concentrations are also below the saturation concentrations below the sulphate penetration and seem to be mostly constant. Thus, the anaerobic methane oxidation does not seem to be lowering it far below the saturation concentration.

**This is only observed for the winter month observation and actually the major reason why we invoke an advective addition of methane. The other methane data are above saturation below the SMT.**

Introduction Line 55: Would be good to put the Tg into perspective to the global flux to know the importance.

**Good point. Will present this a percentage of the total marine methane emission.**

Line 66 and later in the methods: If the methane flux shows high spatial heterogeneity, why do you only measure the flux in one core?
The flux is the average of four core incubations and one diffusive flux measurement. Naturally, this variability is a sampling problem if porewaters are used.
Line 70-1 and 517-9: You also say you do not have data from the ice covered period (line 126)

**Unfortunately, there was a problem with the analysis of the winter data, which we deemed problematic chose not to use.**

Line 91: If you measure over four seasons, I feel seasonal is better than annual. **Okay**

Materials and Methods: Line 102: Do you have info about the CH4, POM and DOM of the effluent of the STP?

**These can be provided, but historically, because of continuous improvement in the sewage treatment, but at the same time rapid growth of the Stockholm metropolitan area, the emissions and C/N/P composition have changed historically. Today's numbers may be misleading to understand the effects for buried sediment carbon from 20 to 30 years ago. For example, in 1994 the treatment plant treated the sewage of 250000 people for the southern Stockholm area. Today, the plant operates at its capacity and treats the sewage of 314000 people + 35000 additional people equivalents from industry ([http://www.syvab.se/himmerfjardsverket/energi-och-materialflode](http://www.syvab.se/himmerfjardsverket/energi-och-materialflode)).**

Line 113: How thick is the rusty brown surface layer?
**It changes significantly from 1 cm to complete absence.**

Line 113-116: Do you have information about the grain size or permeability? That would really be needed to argue for or against advective transport.
**These sediments are fine-grain muds with a small sand fraction. No exact grain size has been performed.**

Line 125-6: What did it mean that there was ice coverage? Line 127:"until for the experiment" change Line 133: "1N HCl" please change to 1M
**Will do.**

Line 134-5: Drying for 2 hours seems short.
Did you test if longer drying had an effect?
**Yes, this works if the oven is at the temperature and no more than 1 g of sediment is used. In operation, the samples are left in the oven until it has cooled to room temperature. Therefore sediments tends to be in the longer for longer than the two hours.**

Line 139: "seconds" seems a bit overstated, especially knowing that it takes already probably more than seconds before the core was on the ship.
"less that a minute" still sounds very impressive and is more realistic.
**We are not talking aboput 10 seconds or less, but less than a minute, ie., tens of seconds. We are really fast with this, because the boat's freebord is low and the core is very quickly brought on deck, but we can change this.**

Line 144: "exactly" delete

**Okay**
Line 145: "5M NaCl" that is not a standard treatment, did you test if it halted microbial activity? Why did you not use base?

**If one calculates the solubility of CH4 in 5M NaCl, CH4 solubility is negligble, and due to the osmotic effect microbial activity likely ends very quickly as well. We do not aagree that this is a non-standard method. There are plenty of publications, which use strong NaCl brine. NaOH is useful if one is interested in the DIC concentration, but in our case we did not process for DIC. Also, the resultant solution is harmless and cheap (kitchen salt) as opposed to strong base. In addition, the exchange of clay-bound $CH_4$ is enhanced in strong salts, which cannot be achieved with 2.5% NaOH.**

Line 146: If you only leave the sample for 1 hour you will not get all the gas adsorbed to clay minerals.
**This is very likely the case in these muddy samples.**

Did you do later measurements to determine if the concentrations were constant?
**Yes, the concentrations remain constant. We have conducted long-term tests, but for the sake of space in the methods description, we chose not to include every aspect.**

Line 149: What column did you use on your GC?
**Porapak Q pre-column (3 feet) with Hayesep D (9 feet), 1ml+1ml sequential loop injection with luerlocked syringe through dried $Na_2SO_4$ filter with quartz wool endings, 60 ml flow rate ($N_2$ 5.0). FID operation with zero air produced, $H_2$ produced with $H_2$ generator. Carrier activated carbon-cleaned cartridges and H2O removal, simultaneous detectors for FID and ECD on hard-cut program.**

Line 162: "10% HCl" is that 10% concentrated HCl in water or is that a 3.7 dilution of concentrated (37%)

**We will change this. 10% of concentrated, i.e., 3.7% HCl** Better give M concentrations.

Line 174: Why did you add cold sulphate to the tracer solution? That introduces sulphate into the sulphate free zone and does not do much in the not sulphate free zone.
**We disagree. Non-amended tracer yields unrealistically high rates in the SMT because of limited sulfate and very high tracer turnover up to 50% of he added tracer, i.e., this would not be a tracer experiment any longer. In order for all incubations to be considered equivalent, a tracer turnover of less than 1% during the incubation is desirable to avoid kinetic limitation due to Michaelis Menten effects, irreversibility effects, etc. An additional beneficial effect is that potential SRR below the SMT can be detected and a cryptic sulfur cycle can be recognized.**

Line 185: "(SO42-)"does not appear in the formula
**Thanks. Will correct this.**

Line 188-189: If you only show the median, what is the error? Plot it in the graph, or if it pretty much constant, state it here.
**The standard error is now reported in the revised figure 5. Replicate measurements are shown in Figure 2.**
Line 215: Why do you use a different fixing agent for the methane samples? Why do you not use base?
**See above**
Line 225-229: The list of variables and the equation do not fit to each other.
**We will adjust the units.**
Line 233-4: Volumetric units do not match.
**Thanks: We will change ppm to nmol**
Line 237-42: Did you see any signs of burrows in your cores? Why do you only do one replicate if you know that it is spatially heterogeneous?
**A: The whole-core incubations are based on core replicates, the methane porewaters are singular cores. Our choice of data is a balance between replication, station time, sample numbers. Almost all core studies on porewater methane in the literature are on singular cores. We have conducted selected replicate experiments, but not for the whole dataset.**
**Bioturbation is seen at Station B1 in the topmost 2 cm as described in the text. These, however, do not pertain to the diffusive flux calculations done for the SMT processes. As far as accounting for the degree of bioturbation is concerned, the resolution of the sampling at the sediment surface is the same as the bioturbation depth. Therefore, there is no possibility to account for bioturbation at the resolution of the sediment sampling.**

Line 248: "Do" why was it recalculated?
**Do is sensitive to temperature and salinity. See Boudreau (1996).**

Results
Line 263: In the abstract you state that methane concentrations exceeded 6mM.
**We will correct to 5.7 mM.**
Line 267-270: it is hard to see linearity or concave shapes in log plots. If these are important don't use log plots.
**The decision in favour of logarithmic plots was primarily because we wanted show the variability of the porewater concentration relative to the saturation concentrations during the different observation periods. In addition, the range of the concentration changes in a core are also very substantial so that the logarithmic scale does better justice to the variability within a core. The saturation limit lines would disappear if the natural range of concentrations were shown. We are following the recommendations by the reviewers, and have changed to a linear scale in addition to showing sulfate and methane data in the same graph, but want to emphasize the advantages of the logarithmic presentation.**

Line 271: You state that the methane profiles can not be explained by the T and Corg changes. How about the sulphate profiles?
**Our analysis indicates that the gradient of sulfate is as much influenced by heterotrophic sulfate reduction as by methane oxidation. The two cannot be separated, and it is therefore not possible to judge on T and org C changes for sulfate reduction alone.**
Line 303: You would also get a rate in the sulphate free zone if you would have injected tracer only. If you use the sulphate concentrations from you profile to determine the sulphate reduction rate? It does not matter that the tracer is reduced. If there is no sulphate the rate is still 0.
**We disagree. There are now a number of publications that demonstrate the existence of the cryptic sulfur cycle (Holmkvist et al., 2014 GCA), but also indications by Leloup**

et al (2009) Environmental Microbiology) that demonstrate the existence of active sulfate reducers below the SMT zone.

Line 305-7: Why is that happening?

**An interpretation is provided in the text? See above.**

Line 313-7: All those rates should not be negative. A negative oxygen uptake of the sediment means diffusion of oxygen out of the sediment. Did you test if you could detect changes in sulphate concentrations in the whole core incubations?

**No, we did not check for changes in sulfate, since the sulfate is vary large compared to the uptake and the precision of a sulfate analysis is no better than 100 μM. By convention, fluxes into the sediment are negative (i.e., oxygen and sulfate, whereas fluxes out are positive, i.e., CH4. This is what is indicated in the tabel and text.**

Line 319: how about reoxidation by oxygen not only iron? Why do you suddenly have reactive iron here, when you state in line 484 that there is no other electron acceptor available?

**Bonaglia et al (2014) give oxygen penetration depths for these sites. These are between 100μm and 0.5 cm. If there is a lack of a sulfate gradient, it is more likely that iron is the intermediate oxidizing agent.**

Line 319: You have to keep in mind that the methane profile does not really give you a rate of methane diffusion out of the sediment. If there is oxygen in the surface sediment there is likely aerobic methane oxidation as you discuss yourself. Thus, this is an over estimation of the methane flux, and you do not know by how much. You need to have methane concentration data at the scale of the oxygen consumption to determine that.

**We agree. Therefore we have in addition conducted whole-core incubations, with the caveats that this method has due to depressurization effects. Publsihed data often use porewater gradients and it is therefore useful to present both types of analysis.**

Line 328-9: As your diffusion based fluxes are overestimated (see comment above) there is no agreement, and thus bioturbation and irrigation, as well as advection as a result of your stirring probably effects the flux.

**The diffusion-based fluxes are not overestimated at depth, where the resolution is sufficient and oxygen plays no role. We will, however, address this aspect in the discussion.**

Discussion

Line 334-7: Do you have data available to support that? Can you model that it does not fit?

**This was a statistical test that showed the lack of correlation. We do not think that a model would provide a satisfying or sufficently reliable answer to this question.**

Line 342: How about rate studies in temperature gradient blocks, e.g. by Sagemann or Arnosti?

**We have unpublished temperature gradient block data on SRR for Himmerfjärden sediment. These support the statement in text.**

Line 348: How big could the effect be with this difference in salinity?

**It is not relevant, mostly because sulfur cycling in the topmost cm makes sulfate multiple times available far in excess of concentrations variations due to salinity changes.**

Line 352: How much Corg comes from the sewage treatment plant?

**The sewage treatment emits minor amount of POM and DOM compared to the nutrients nitrate and phosphate, which stimulate plankton production in the estuary (Bonaglia et al., 2014). It is the nutrient effect that is most relevant for carbon. Carbon estimates of the contribution by the sewage treatment plant were done by Savage et al.**

How similar or different is the fjord thus to others?

**The inner parts of the fjord share similarities with, e.g., Oslo fjord, whereas the outer prarts are quite pristine and may be comparable to other northern latitude fjord-type systems. However, many fjord systems have significantly higher salinities so that sulfate reduction pervails over a thicker sediment layer than in these sediments.**

**Another difference is the glacial history of the Baltic Sea sediment. During this time organic-poor lake sediments were deposited. In other fjord sediments, this discontinuity to a freshwater phase may not exist in the geological record. This will affect the methane generation potential and thereby the methane flux.**

Line 357-9: Where do the high sedimentation rates come from if there is only low river runoff?
**These sediments are accumulation bottom sediments, which have a significant proportion of redeposited fine-grained material that is transported laterally and deposited in the bathymetric depressions of the fjärd.**
Line 385: Table 1 has no information about the burial of organic material. Do you have depth profiles supporting that?
**Published organic carbon concentration profiles can be found in Thang et al. (2013). At Station B1**
Line 391: Salinity of B1 is 7‰.

Line 392-4: A little too often "compar*"

Line 392-8:
Lower sulphate concentrations mean that there is less sulphate available for organ- oclastic & methanotrophic sulphate reduction, just by simple numbers.
**We do not dispute this.**

Line 410&2: Figure 4

Line 412-3: Iron and manganese reducers do not always outcompete sulphate reducers, see work by Thamdrup and Vandieken.

**Yes, but in these sediments this is the case. We have conducted bag incubation and iron and Mn speciation analysis in bag incubations in core profiles to 10 cm depth. These data indicate that BSR account for 75% of organic matter oxidation, Fe reduction about 6.5 % and Mn reduction about 2.5%. The rest is accounted for by heterotrophic denitrification (Goldschmidt presentation Downs and Bruchert (2013); Bonaglia et al. (2014) Biogeochemistry).**

Line 414-5: You state that the main driver for the differences is the advective flow based on the hydrostatic pressure, but here you speculate about more sulphate reduction leads to less methanogenesis. Which process is now the important one?
**As said above, it is not one OR the other, but an interplay of multiple processes with varying influences on the system. This is also why these sediments would be extremely hard to model accurately in one-dimensional reaction-transport models.**

Line 417-20: How deep is the bioturbation in these sediments?
**An exact bioturbation depth would be arbitrary. Macrofauna analysis at H6 and B1 has shown that Marenzelleria does generally not go deeper than about 4 cm, but can occur occasionally down to 10 cm.**
Line 419-21: In advective systems with bioturbation fluxes should increase not decrease.

**We disagree. Since more oxygen can be imported, it is possible that methane oxidation increases.**

Line 443-5: This is not an explanation, it is just stating that you believe the data in contrast to the scenario below.

**Possible explanations are provided in the following lines. 444f**

Line 454: "law" replace with "function" Line 461-3: Sentence not clear Line 465: What is the percentage if you compare the methane flux into the SMTZ with the accumulated SRR or the total methane flux with the SRR? Do the numbers fit what model says?

**We can replace 'law' with function, but the term has been used in the literature sicnen Jorgensen (1979) and also in Jorgensen and Parkes (2010). Lines 467f state what the reviewer asks for, i.e., the depth-integrated SRR relative to methane flux rates fit well with the model.**

Line 474-7: In line 445 you state that there is only little link.

**I am sorry but we do not see the apparent contradiction the reviewer states.**

Line 480-3: Please provide some support for this, maybe with a model.

**It is the data that showed an abrupt decrease in porewater concentrations for which an explanation is required. The above discussion intended to lay out that neither temperature, salinity nor changes in organic matter influx alone can explain the change. The sentence does not intend to say more than that. It is beyond the scope of the paper to develop a unifying model that can address these processes satisfactorily. Even some of the currently most complete models, such as by Mogollon et al. (2011) JGR are idealizations that may not yield satisfactory fits with our data, but that does not necessarily dispute either model or data.**

Line 484: Did you determine the concentrations of other electron acceptors like Fe? In line 319 you state that it is available for sulphide reoxidation.

**Fe data are available from the nearby Station H5 and published in Thang et al. (2013). In addition, there an unpublished data for nearby Stations H3 and H2 that indicate the limitation of reactive iron in the postglacial mud. In the glacial lake clays, however, reactive iron is more abundant again, but these latter sediments do not control the methane production.**
Line 490-2: Do you have any data on changing pressures? What would drive these changes? What would the possible magnitude be?
**We have tried to obtain water level data and air pressure data for the periods of observation at the sampling stations, but these were not archived or could be found for the precise localities, only for the open Baltic nearby. However, local data is what was needed to have an accurate idea of the hydrostatic pressure.**
Line 500-1: Do you have any data or reference to support this magnitude?
**The best reference to address this question is the study by Mogollon et al (2011 JGR Biogeosciences), who modelled the free gas depth and AOM rates for two stations in southwestern Baltic Sea sediment. In that study temperature, and not tidally influenced pressure change, were found to be the dominant regulators of the free gas depth variation. Our differing intepretation is based on the observation that the seasonal variability in temperature at the two stites studied there are much greater than the ones studied here.**

Line 500: "may as much as: : :" insert "be"
**Done**
Line 497-503: If it is the hydrostatic pressure it is actually not really a seasonal effect?
**Yes**
Conclusion
Line 508-9: Sentence not clear.
**Will revise this sentence to make it clearer**
Line 515-6: How is it seasonal/annual if it is the hydrostatic pressure? Not clear.

Line 517-9: You also state that you do not have data from the ice covered times!
**This must be a misunderstanding, since we have the February data. This is one of the few study that present data at ice cover.**
Line 519-20: Why is that if the temperature and Corg input do not play an important role?
**We emphasize that there are two aspects to be considered. Our winter and early spring data give low rates. Generally, in the literature, there are very few data available for sites with measurements during ice cover. If late spring/summer/fall rates are therefore extrapolated over a whole year, overestimates can result. Models, however, may account for this effect, if they are able to parameterize temperature and reactive organic carbon correctly. Secondly, while our observations indicate that hydrostatic changes or changes in porewater advection may have a considerable influence, this still does not take away from the fact that there is also seasonal variability. To model the annual variability based on organic carbon and temperature alone may therefore unfortunately also give the wrong results, because these factors have not been acounted to a degree that may reflect the specific regional situation. These two aspects need to be considered separately.**

Line 520-1: Is it no advective or diffusive transport that controls the methane? You keep changing your argument.
**Please see the comment above.**

Tables
Table 2: "no AOM zone3" for H6 January 2013, change to "no AOM zone4"
**Will correct this**
Table 3: "Exponential coefficient (a)" this one is not exponential.
**Will correct this**
Figures Make all the y axis for depth the same scale.
**Will correct this.**
Figure 1. The colors do not help reading the map too much, not very clear in black and white at all. Change continent to just white or black to make ocean more clear and maybe reduce shades in the water.
**We will consider a black and white map, or a map with sufficient contrast, clear legends and geographic locations.**
Figure 2: You use the maximum sulphate penetration from figure 3 here. Maybe change the order of the figure to keep the flow consistent.
**We will do that and then consequently also change the sequence in the results description.**
Figure 3: How do you define your maximum sulphate penetration if there is a sulphate peak in your graphs below it?
**Sulfate penetration depth was defined as the first lowest concentration measured. It is a common observation that traces of sulfate in the range of concentrations 50µM to 400 µM remain detectable in porewaters to substantial depths of several meters far into the methanogenic zone. This observation has been addressed in detail by Roey et al., Holmkvist et al and is part of the argument in favour of a cryptic sulfur cycle, which the authors of this manuscript also agree to. We also observe non-zero sulfate concentrations in the methane-rich zones of our sediment cores. We have therefore chosen to define the sulfate penetration depth as that at which the sulfate changes sharply and the sulfate concentration is below 0.5 mM. We think that this combined criterion best reflects the condition of a transition from a dominating sulfate reduction environment to an environment in which methane-cycling processes start to prevail (while acknowledging low rates of cyptic bacterial sulfate reduction).**

Figure 5: Keep order consistent between listing in the graph and in the caption. If you keep all the values positive it is much easier to compare the values. Which method are the methane fluxes based on?

**We use the convention from the perspective of the water column, i.e., loss fluxes are fluxes into the sediment and negative and gain fluxes into the water column are positive.**

---

## Author Response (AR1)

**Author's response**

**"Annual variability and regulation of methane and sulfate fluxes in Baltic Sea estuarine sediments" by Joanna E. Sawicka and Volker Brüchert**

**Anonymous Referee #1**

1.) General Comments

The paper investigates the effects of changes in temperature, benthic oxygen concentration and eutrophication on the sediment concentrations and fluxes of methane (and sulfate) in two sites in the Baltic Sea, an open-water coastal site and a eutrophic estuarine site over 4 time points (spring, summer, fall and winter) within a 12-month season. In order to address this, the authors measured methane and sulfate concentrations, oxygen uptake and sulfate reduction rates and calculated sulfate and methane fluxes in the sediment in the upper approx. 45 cm of the sediments. The main influence on methane emission from the sediment was found to be by bottom water oxygen enhancing aerobic carbon mineralization and oxidative recycling of sulfate. The authors state that the seasonal changes in sediment methane concentrations are too large to be only the result of changes in methane generation and oxidation. Thus, they suggest advective recharge of methane from deeper, gas-rich sediment layers as possible influencing factor. The methane concentration below the sediment surface is lowered by AOM below the saturation concentration and thus bubble emission does not play a role at the investigated sites. The study presents a well-designed experimental set-up and the experiments are performed thoroughly.

However, the authors fail to formulate a clear scientific objective to conduct this research. It should be clear from the abstract and from the introduction why this study was conducted and what the expected merit would be. The abstract describes the findings and ends with the conclusion but it does not clearly mention the scientific questions addressed. At the beginning, the importance of this study should be made clear to attract the reader attention and interest, e.g. by naming the research question behind. Such questions can then be answered by the findings.

The introduction is well written, describes the state of the art and highlights some gaps in knowledge to justify the study. It also briefly summarizes the methods applied in the study. However, a concise statement what the presented study will contribute would make the paper sound much stronger. As mentioned above, it would be good if the authors state what problem they exactly address and how they do it – in other words, what exactly do they want to find out by the applied methods.

**Answer: We reformulated our objectives and scientific aims and emphasized the scope of the study somewhat clearer.**

The presentation of the results is confusing. There is major work needed to check the consistency of the figures and the text (see specific comments below). This makes it hard to follow the argumentation as one cannot relate the described results to the profiles. When presenting the results, I would suggest sticking to the same order of the stations throughout the entire manuscript. For example, always describe station B1 first and then station H6 and have the same order also in the tables and figures (i.e., B1 on top and H6 below). The whole section should be rewritten with a focus to guide the reader clearly through the graphs. More attention should be paid to the general consistency in the style of units. For example: mM vs. mmol/L vs. mmol L-1 (or e.g. nmol cc-3 d-1 and mmol L-1 OR nmol/cm3/d and mmol/L).

**Answer: We have followed these suggestions to improve the clarity. See editions in Results section.**

I suggest combining Figure 2 and 3 by plotting $CH_4$ and sulfate concentration in the same plot with linear concentration scale also for methane. The logarithmic scale for methane makes it hard to follow the changes and it is easily to compare with the sulfate profile if both are on the same scale and together. I also suggest showing all data of the triplicates for the sulfate reduction rates in Figure 4 and making the fit - not only from the medians (see details below).

**Answer: We followed the recommendation and combined Figures 2 and 3..**

The interpretation and argumentation as well as the conclusions seem reasonable and are well written. The conclusion contains many good arguments and statements of which I think it would be good to mention these in the abstract to raise the interest of the reader.

I suggest publication of this interesting study in Biogeosciences, however, I indicated major revision because the results presentation needs some careful rewriting with better guidance for the reader as well as careful cross-checking of text and figure/table content.

I think it is worth to add than 3 key words, to help finding the paper.
**Keywords: Methane cycling, coastal and estuarine sediment, seasonality**

2.) Specific Comments

Lines 30-32: rephrase the sentence, and maybe split. At the moment it says that "The effects of temperature [: : :] where **(A: where ?)** investigated [: : :] for open-water coastal and [: : :] sediment." That is probably not what the authors wanted to say.
**Answer: We simplified this sentence, but we are not exactly sure what the reviewer meant by this comment.**

Line 68: I would delete "summer"
**Done**

Figure 1: is of rather bad quality (at least in the document I could print out). It is impossible to read the names of rivers, cities or islands. Maybe the colors could also have more contrast to make the whole picture look sharper. A color code/legend could be helpful to understand the different blue tones (is this water depth?). If this differentiation is not important, a single color for water would be better.
**Thanks for the suggestion. We have produced an alternative map in Ocean Data View, however, without the rastered high-resolution bathymetry. We would have liked to retain the bathymetric information, because it is important to understand the sediment deposition pattern, but cannot provide it at an overall satisfying resolution, because the bathymetric data are rastered for a large coastal area and cannot be presented as high-resolution cutouts for Himmerfjärden.**

Lines 127, 129, 133, 148/149 : name equipment manufacture here
**Okay. Done**
Lines 180 – 185: total reduced inorganic sulfur should be abbreviated with TRIS, at least it should be consistent with the formula.
**Okay. Done**

Line 189: I think it is better not to use the median here. Out of three measurements (triplicates) the median will always be the measurement in the middle. This means your plot and the input for the mathematical fit only relays on the one measured result (although there is the information behind that there is one higher and one lower measurement). You might talk about mean values in the text but in general I think you should present all individual measurements and also plot all data in the plots in Figure 5. And then you can calculate a fitted curve and also include this in the plot to visualize trends. It might be that the individual measurements show outliers and individual replicates differ. However, this is not uncommon for rate measurements and the best strategy is to simply show all data. Otherwise it could make the impression that something was tried to hide behind the median.
**A: The format was chosen to present the trends in the best possible way. We have revamped the figures and now show all data points.**

Line 245/256: be consistent in the order of described results (e.g. B1 as first and H6 as second) within the text for all parameters and also with table 1.
**We carefully checked and revised the text to be consistent in the sequence the stations are listed.**

Line 262ff: Please indicate the individual figure numbers after each station and result, e.g., "August... at station H6 (Fig. 2f) and : : :at station B1 (Fig. 2b) and so on for all mentioned data, this helps to identify the results in the figures. Please also make the order consistent over the entire manuscript. Moreover, here are some inconsistencies between text and figure that could be easily sorted out by referencing to the respective profile. For example, in the text is says highest $CH_4$ concentrations in August (H6: 5.7 mM, B1 1.9 mM). While for Station H6 (Fig. 2f) this might be true, for station B1 (Fig 2b) the figure I cannot see the 1.9 mM, in fact August 2012 has the lowest methane concentrations. Also in February, only at H6 (Fig, 2h) the $CH_4$ concentration is lowest, but not at B1 (Fig. 2d). However, the number mentioned in the Text for B1 Feb 2103 (0.1 mM max.) matches the highest values in the B1 Feb 2013 figure (Fig. 2d) but it is not the lowest $CH_4$ concentration in B1, this is in August 2012 (Fig. 2b).
**Thanks for this suggestion. We adapted the changes to the text accordingly.**

I am wondering about the use of a logarithmic scale for the methane concentrations, this is unusual for the presentation of sediment methane concentrations. The mentioned linear increase in sediments at H6 is not visible due to the logarithmic concentration scale and also not the described "concave upwards trend" for B1. Also the mentioned differences in maximum concentrations are not visible due to that scale. Here, a linear concentration scale would be better to visualize the concentration changes. It would furthermore allow for a better judgement of the data quality and the efficiency of the sampling protocol (in terms of potential methane loss). A linear scale would also be helpful to compare the data with the sulfate data and the maximum sulfate penetration depth indicated by the green line. When using the linear scale it could be a good idea to combine Methane and Sulfate Profiles (Figures 2 and 3) in one plot for each sampling point.

**A: We have reconverted the figures back to a linear scale. We were well aware of the potential criticism the use of a logarithmic scale can draw. However, logarithmic scales have been used frequently, e.g., in the IODP literature. Here the main purpose was to convey the scale of the concentration change and to demonstrate that the sampling technique allows us to capture concentrations above the concentration at atmospheric pressure with consistency, but that the measured concentrations were consistently far below the saturation concentration at in situ pressure. The linear scale does not allow us to do this. However, we welcome the honest opinion of the reviewer, and therefore reverted to the conventional linear scale.**

Line 269: "concave upwards trend" what is meant by this? This is very unusual for a profile description. Do you mean increase followed by decrease? Here also a linear concentration scale would help to understand.

**A: Scale was changed back to linear to avoid more confusion.**

Line 272ff: I do not see that the sulfate concentration gradient at station H6 in October 2012 (Fig. 3g) . For me it seems that the steepest increase is in August (Fig.3f) (>6 mM over < 10 cm depth)

**A: The steepest gradient occurs between 6 and 10 cm depth in October whereas the surface gradient is less steep. This is likely due to the increased $O_2$ concentrations and the colder temperature in the fall leading to a downward propagation of a less steep sulfate gradient, however, not yet at a depth below 6 cm.**

Line 275: better : "At station H6, sulfate was always fully depleted within the cored sediment interval, : : :"
**A: okay, changed**

Line 276: "Depletion already occurred at 5 cm depth in April and October and at 9 cm in August: : :" Depletion occurs all the way down from the surface sediment to the lowest concentration in the profile. Do you mean complete depletion (or depletion until a low constant level)? This is at approx. 9 cm depth at H6 in August 2012 (Fig. 3f) but I cannot see the 5 cm in April (Fig. 3e) and October (Fig. 3g), or do you refer here to Station B1 (Fig. 3a-d)?
**A: The reviewer is correct. We changed this.**

Figures: 2-3 It would be helpful to quickly identify the profiles mentioned in the text, if station number are indicated, e.g., for each row. The letters (a), (b), etc., should be larger in order to better overview the figure and relate it to the text while reading. The style of the units should be consistent with the format used in the text (mmol/L vs. mM). As mentioned in the comments above, I suggest combining figures 2 and 3 and presenting the methane concentration with a linear scale.
**A: We have increased the numbers and assign the station names to each row. We also plot sulfate and methane together in one plot, Figure 2**

Line 287 ff: also here, please indicate the related profile in the figure 4 always directly when mentioned in the text, to help the reader understanding the text quickly.
**A: Done**

Line 288/289: I don't see an SRR increase to 63 nmol cm-3 d-1 in any of the profile of B1. The maximum SRR I see is in Fig. 4a at approx. 35 nmol cm-3 d-1. Also for station H6, I do not find a maximum of 411 nmol cm-3 d-1 in the figure. The maximum measured is around 350 nmol cm-3 d-1 in Fig. 4e. Are these the individual measurements (i.e. from one of the triplicates?) As mentioned above, I suggest showing the data of all triplicate samples. If you refer to theoretical values at the very surface calculated from the regression line, please indicate so.

**A: We now show all replicates and show the regression lines based on the power law model. The author is correct in that assuming that the reference in the text was made for the individual measurements and not the averages.**

Line 305: What is the "peak between 6 and 9 cm depth? Isn't that a second peak? Sulfate is already mostly depleted at 10 cm and CH4 seems to be at maximum concentration below 10 cm. Could this increase SRR her not indicate AOM? Again, an overlay of sulfate and methane concentrations profile with linear concentration scale (combined Figs 2 and 3) would help to judge this better.

**A: We had of course also debated the second peak. It is a consistent feature, since it is also nicely visible in the 3 replicate SRR. However, there is no change in the sulfate gradient or the methane gradient in this depth intervals that would be expected if the measured rates would be largely attributable to AOM. Therefore, we consider it less likely that the SRR at this depth indicate largely AOM and interpret this peak as due to organiclastic SR.**

Figure 4: please indicate what H6 and B1, in the caption and best also in the Figure itself (e.g. for each line of plots). As mentioned earlier, I would like to see all individual data here instead of the median.
**A: Done.**

Figure 5: A separation line between the four sampling times would be helpful for a better readability.! Maybe also indicate them with Letters and reference to the plots in the text when described. Add the error bars if the errors are mentioned in the text. The figure says February 2013 but the Table January 2013 Line 314/315 and Fig 5: In the fig 5, highest TOU at H6 is in April (_33 mmol m2 d-1) and at B1 in August _22 nmol cm-2 d-1) or so, contradictory to the text.

**A: We have changed the figure accordingly and adjusted the text.**

Line 315: sulfate flux seems to be lowest at B1 in August not in February and highest in February or April, contradictory to the text.
**A: We assume that the reviewer refers to the difference in uptake calculated for the SO$_4$ gradient and the SRR for August 2012. We have clarified in the new Figure 4 that the sulfate flux data are the $^{35}$S data.**

Line 398/399 ": : :constraints decide on the result of this competition between these two processes."
**A: Done**

3.) Technical Corrections
line 144: cut-off
**corrected**
line 147: replace "to force out" by "to push"
**revised**
line 149: "CH4 standards at 100 ppm and : : :"
**inserted 'of'**
line 156: (cm3) instead of (cubic centimeter)
**changed**
line 162: missing dot after et al.
**added**
line 167: missing word after "adjacent"
**inserted 'intervals'**
line 177: 1 cm intervals line 196: 40 L incubation tank line 218: mL ("L" consistent to previous use)

**changed**
line 221: replace "to force out" by "to push"
**changed**
line 223: "CH4 standards at: : :"
**'of'**
line 227/228: 0.003 L / 0.009 L
**inserted the space**
line 257: 300 _M
**µM**

line 467/468: remove one "integrated" in the sentence.

**Removed the first 'integrated'**

Review "Annual Variability and regulation of methane in sulphate fluxes in Baltic Sea estuarine sediments." by Joanna E. Sawicka and Volker Bruechert

Sawicka and Bruechert study the seasonality of methane flux and sulphate reduction in two coastal sites in Sweden. With estuaries being important players in the global methane cycle, it is important to gain more insight into the controlling factors of methane oxidation in these systems.

Major comments:
There are several assumptions in the manuscript that are not backed up by either data or references.
E.g.
Line 47-48: Importance of advective processes.

**See comments below**

Line 333-337: Temperature regulation inference.

**A: Sole temperature regulation would imply that methane oxidation is less temperature-sensitive than methanogenesis preventing methane oxidizers from keeping up with the enhanced methane flux during summer. This requires significantly different values of $Q_{10}$ of methanogens and methane oxidizers. Publications from lake environments and terrestrial environments, e.g., King (1992), Wik et al. (2014), Nguyen et al (2011) suggest that aerobic methane-oxidizing bacteria may have higher $Q_{10}$ than methanogens, but this argument remains unproven for marine habitats. In case of anaerobic methane oxidation, it is difficult to argue for a temperature adaptation disadvantage of AOM compared to methanogenesis, because of the tight coupling between sulfate reduction and methane oxidation and the phylogenetic proximity w/ respect to 16S of ANME to known methanogens, but also with regard to Archaeal membrane lipid composition, which should be considered strong physiological regulator for cross-membrane transport.**

Line 348 salinity variation.

**A: Data shown in Table 1 indicate that the salinity for the different sampling periods varied little.**

Line 480-483 Variability in methane concentrations not due to variability in methane oxidation rates alone

**A: The emphasis here is on the word alone. The whole first paragraph of the discussion emphasizes the different regulatory processes that affect methane concentrations and two important ones are of course temperature and bottom water oxygen.**

Line 490-492: Changes in the upward transport rate of methane

**A: Methanogenesis rates can only increase due a temperature increase, since the availability of organic carbon for methane production in buried sediment does not change. These effects were modelled by Dale et al. (2006) and are discussed here.**

Line 498-501. Migration of the methane saturation zone due hydrostatic pressure changes

**A: There is acoustic echosounder evidence for free gas presence in these sediments and the authors have personal communcation (Tom Floden) evidence that the depth of the free gas zone as seen on the acoustic echosounder changes substantially from year to year. The mechanisms that affect gas migration in these sediment are manifold. They can have to do with atmospheric pressure changes, wind direction or affecting water levels and these in turn affect the solubility of methane at a given temperature. An additional parameter is groundwater movement. There is also geophysical evidence from other areas in the outer Himmerfjärden area that suggest groundwater seepage. This implies a complex aquifer hydrology that was not accessible with the coring methods used here, but that has indirect effects on methane solubility, advective transport, and effective methane flux. A mechanistic or model-based evaluation of all these processes is far beyond the scope of this paper and cannot be adequately addressed in addition to the data presented here. We have explained this in lines 547-553 in the text.**

Line 519-520. The period of ice cover has low flux rates. Extrapolation of rates during open-water conditions for a whole year would therefore be overestimates.

**A: Yes, we agree. To our knowledge, this is one of the very few studies that reports sulfate reduction rates in a fully ice-covered estuary. These rates were very low compared to the open-water season. Extrapolation of these rates for the ice-covered period will necessarily give lower annually integrated rates compared to rates based on open-water measurements in spring/summer/fall. During certain years, the ice-covered season lasts from late December to early April. Ignoring winter data for extrapolations would give very erroneous numbers.**

The authors go back and forth about stating if the methane transport is controlled by diffusion or advection. Sediment permeability would help to understand what role advection can play.

**A: We do not have measurements of permeability available. While this could help, we would like to point out that even low-permeability sediment emit bubbles.**

They state that changes in the hydrostatic pressure drives the changes in methane profiles, but do not explain what drives the changes in hydrostatic pressure, nor if that is related to season or not.

**A: Changes in hydrostatic pressure in the Baltic Sea are influenced by air pressure, prevailing wind direction, and general sealevel stand due to the balancing effects of saltwater entry through the Danish straits and freshwater discharge in the northern Baltic, and from the rivers flowing in from the south. Additional effects are caused by the local coastal topography. These multiple parameters result in complex subsurface hydrology and complexity in estuarine water level conditions that make it difficult to use general meteorological observations to predict local sealevel variability. Hydrographic data are only reported for the general area in open water at the Landsort island, but these are not the same as in the archipelago.**

A lot of equations are listed that are just taken from other publications. Those do not need to be listed again.

**A: We have chosen to retain the formulas. There are enough readers that are unfamiliar with some of the methods. Without going into too much detail, these equations provide the basic framework.**

It helps reading the manuscript if you keep the order of things the same, best throughout the manuscript (e.g. first mention station B1 then H6) but definitely in the same or consecutive sentences (e.g. Line 185-186, line 287 to 291). The order changes frequently in the manuscript making it harder to follow the arguments.

**A: We have revised the manuscript to make sure the sequence is adhered to consistently.**

Minor comments
Title "Annual variability : : :" if it is mostly the pressure it seems that the year plays not an important role here, so I think annual is not very good. If it is the seasons, I think seasonal is better. ": : : Baltic Sea estuarine: : :" but you say later that you investigated an estuarine and an open water station. Better say coastal?

**A: Thanks for pointing this out. We changed this.**

Abstract Line 41: You list 5.7mM as max in line 263 Line 43-4: ": : :lowering: : : far below the saturation concentrations." Your methane concentrations are also below the saturation concentrations below the sulphate penetration and seem to be mostly constant. Thus, the anaerobic methane oxidation does not seem to be lowering it far below the saturation concentration.

**A: This is only observed for the winter observation and actually the major reason why we invoke an advective addition of methane. The other methane data are above saturation below the SMT.**

Introduction Line 55: Would be good to put the Tg into perspective to the global flux to know the importance.

**A: We have added data on the proportion of coastal emissions and compare them to the total estimated marine methane emission. However, the choice of a good number is difficult, partly because of the ongoing debate on the contribution of coastal methane emissions from thawing permafrost in the Siberian Arctic and uncertainties in correctly assessing macro/micro-seep-related emissions.**

Line 66 and later in the methods: If the methane flux shows high spatial heterogeneity, why do you only measure the flux in one core?
The flux is the average of four core incubations and one diffusive flux measurement. Naturally, this variability is a sampling problem if porewaters are used.
Line 70-1 and 517-9: You also say you do not have data from the ice covered period (line 126)

**A: This must be a misunderstanding, because we collected samples in late January/February with a ship capable of breaking the coastal ice.**

Line 91: If you measure over four seasons, I feel seasonal is better than annual.

**A: We changed this.**

Materials and Methods: Line 102: Do you have info about the CH4, POM and DOM of the effluent of the STP?

**A: We have provided reported numbers on the treatment plant emissions for 2012. We would like to point out, however, that because of continuous improvement in the sewage treatment operation since the 80's, but at the same time the rapid growth of the Stockholm metropolitan area, the emissions and C/N/P composition have changed historically. Today's numbers may be misleading to understand the effects for buried sediment carbon from 20 to 30 years ago and only N and P data are available for the long period. For example, in 1994 the treatment plant treated the sewage of 250000 people for the southern Stockholm area. Today, the plant operates at its capacity and treats the sewage of 314000 people + 35000 additional people equivalents from industry ([http://www.syvab.se/himmerfjardsverket/energi-och-materialflode](http://www.syvab.se/himmerfjardsverket/energi-och-materialflode)), but N and P emissions are at pre-1970's levels.**

Line 113: How thick is the rusty brown surface layer?

**A: It changes significantly from 1 cm to complete absence.**

Line 113-116: Do you have information about the grain size or permeability? That would really be needed to argue for or against advective transport.

**A: These sediments are fine-grain muds with a small sand fraction. No exact grain size has been performed.**

Line 125-6: What did it mean that there was ice coverage? Line 127:"until for the experiment" change Line 133: "1N HCl" please change to 1M

**A: Changed.**

Line 134-5: Drying for 2 hours seems short.
Did you test if longer drying had an effect?

**A: As a matter of fact, the actual period was longer than that since the samples were retained in the oven. The calculated porosities are consistent with measurement from nearby stations.**

Line 139: "seconds" seems a bit overstated, especially knowing that it takes already probably more than seconds before the core was on the ship.
"less that a minute" still sounds very impressive and is more realistic.

**A: Less than a minute passes before we start removing samples. To avoid doubt, we have changed this to minutes since the whole procedure takes about 5 minutes for a Multicore.**

Line 144: "exactly" delete

**A: Changed**

Line 145: "5M NaCl" that is not a standard treatment, did you test if it halted microbial activity?  Why did you not use base?

**A: If one calculates the solubility of $CH_4$ in 5M NaCl, $CH_4$ solubility is negligible, and due to the osmotic effect microbial activity likely ends very quickly as well. We do not agree that this is a non-standard method. There are plenty of publications, which use strong NaCl brine. NaOH is useful, if one is interested in the DIC concentration, but in our case we did not process for DIC. Also, the resultant solution is harmless and**

**cheap (kitchen salt) as opposed to strong base. In addition, the exchange of clay-bound CH₄ is enhanced in strong salts, which cannot be achieved with 2.5% NaOH.**

Line 146: If you only leave the sample for 1 hour you will not get all the gas adsorbed to clay minerals.

**A: This is very likely the case in these muddy samples.**

Did you do later measurements to determine if the concentrations were constant?

**A: Yes, the concentrations remain constant. We have conducted long-term tests, but for the sake of space in the methods description, we chose not to include every aspect.**

Line 149: What column did you use on your GC?

**A: The GC is equipped with a Porapak Q pre-column (3 feet) followed by a Hayesep D (9 feet) analytical column. We use two sequential 1ml+1ml loops for injection with luerlocked glass syringe. Injection occurs through a dried $Na_2SO_4$ filter bed with quartz wool endings. Flow rate of the GC is 60 ml flow rate ($N_2$ 5.0). The FID operation occurs with zero air produced clean over a carbon filter and $H_2$ is produced with a Schmidlin $H_2$ generator. Gas cleanup occurs via activated carbon-cleaned cartridges. More details have been added to the method description.**

Line 162: "10% HCl" is that 10% concentrated HCl in water or is that a 3.7 dilution of concentrated (37%)

**A: We changed this to 1 M HCl.**

Better give M concentrations.

Line 174: Why did you add cold sulphate to the tracer solution? That introduces sulphate into the sulphate free zone and does not do much in the not sulphate free zone.

**A: We disagree. Non-amended tracer yields unrealistically high rates in the SMT because of limited sulfate and very high tracer turnover up to 50% of he added tracer, i.e., this would not be a tracer experiment any longer. In order for all incubations to be considered equivalent, a tracer turnover of less than 1% during the incubation is desirable to avoid kinetic limitation due to Michaelis Menten effects, irreversibility effects, etc. An additional beneficial effect is that potential SRR below the SMT can be detected and a cryptic sulfur cycle can be recognized.**

Line 185: "(SO42-)"does not appear in the formula

**A: Thanks. We corrected this.**

Line 188-189: If you only show the median, what is the error? Plot it in the graph, or if it pretty much constant, state it here.

**A: The standard error is now reported in the revised Figure 4. Replicate measurements are shown in Figure 3.**

Line 215: Why do you use a different fixing agent for the methane samples? Why do you not use base?

**A: See above.**

Line 225-229: The list of variables and the equation do not fit to each other.

**A: We adjusted the units.**

Line 233-4: Volumetric units do not match.

**A: Thanks: We changed ppm to nmol.**

Line 237-42: Did you see any signs of burrows in your cores? Why do you only do one replicate if you know that it is spatially heterogeneous?

**A: Marenzelleria is the dominant bioturbator in these sediments, but does not play a large role for CH4 emissions (Bonaglia et al., 2013 MEPS). Microelectrode profiling (Bonaglia et al., 2014) has not shown distinct subsurface increase due to pumped $O_2$ from burrowing or bioirrigating animals. Bioturbation is seen at Station B1 in the topmost 2 cm as described in the text. These, however, do not pertain to the diffusive flux calculations done for the SMT processes. As far as accounting for the degree of bioturbation is concerned, the resolution of the sampling at the sediment surface is the same as the bioturbation depth. Therefore, there is no possibility to account for bioturbation at the resolution of the sediment sampling.**

**A: The whole-core incubations are based on core replicates, the methane porewaters are singular cores. Our choice of data is a balance between replication, station time, sample numbers. Almost all core studies on porewater methane in the literature are on singular cores. We have conducted selected replicate experiments, but not for the whole dataset.**

Line 248: "Do" why was it recalculated?
**A: $D^o$ is sensitive to temperature and salinity. See Boudreau (1996).**

Results
Line 263: In the abstract you state that methane concentrations exceeded 6mM.

**A: We corrected to 5.7 mM.**

Line 267-270: it is hard to see linearity or concave shapes in log plots. If these are important don't use log plots.

**A: We have followed the recommendations by the reviewers, and have changed to a linear scale in addition to showing sulfate and methane data in the same graph (Figure 2a-h). The decision in favor of logarithmic plots was primarily because we wanted show the variability of the pore water concentration relative to the saturation concentrations during the different observation periods. In addition, the range of the concentration changes in a core is very substantial so that a logarithmic scale does better justice to the variability within a core (see ODP/IODP results). The in–situ saturation limit lines disappear if the natural range of concentrations were shown.**

Line 271: You state that the methane profiles cannot be explained by the T and Corg changes. How about the sulphate profiles?

**A: Our analysis indicates that the gradient of sulfate is as much influenced by heterotrophic sulfate reduction as by methane oxidation. The two cannot be**

**separated, and it is therefore not possible to judge on T and org C changes for sulfate reduction alone.**

Line 303: You would also get a rate in the sulphate free zone if you would have injected tracer only. If you use the sulphate concentrations from you profile to determine the sulphate reduction rate? It does not matter that the tracer is reduced. If there is no sulphate the rate is still 0.

**A: We disagree. There are now a number of publications that demonstrate the existence of the cryptic sulfur cycle (Holmkvist et al., 2014 GCA), and also indications by e.g., Leloup et al (2009) Environmental Microbiology) that demonstrate the existence of active sulfate reducers below the SMT zone.**

Line 305-7: Why is that happening?

**A: An interpretation is provided in the text? See above.**

Line 313-7: All those rates should not be negative. A negative oxygen uptake of the sediment means diffusion of oxygen out of the sediment. Did you test if you could detect changes in sulphate concentrations in the whole core incubations?

**A: No, we did not check for changes in sulfate, since the sulfate concentration is large compared to the uptake and the precision of a sulfate analysis is no better than 100 µM. By convention, fluxes into the sediment are negative (i.e., oxygen and sulfate, whereas fluxes out are positive, i.e., $CH_4$. This is what is indicated in the table and text.**

Line 319: how about reoxidation by oxygen not only iron? Why do you suddenly have reactive iron here, when you state in line 484 that there is no other electron acceptor available?

**A: Bonaglia et al (2014) give oxygen penetration depths for these sites. These are between 100µm and 0.5 cm. If there is a lack of a sulfate gradient, it is more likely that iron is the intermediate oxidizing agent.**

Line 319: You have to keep in mind that the methane profile does not really give you a rate of methane diffusion out of the sediment. If there is oxygen in the surface sediment there is likely aerobic methane oxidation as you discuss yourself. Thus, this is an over estimation of the methane flux, and you do not know by how much. You need to have methane concentration data at the scale of the oxygen consumption to determine that.

**A: We agree. Therefore we have in addition conducted whole-core incubations, with the caveats that this method has due to depressurization effects. Published data often use porewater gradients and it is therefore useful to present both types of analyses.**

Line 328-9: As your diffusion based fluxes are overestimated (see comment above) there is no agreement, and thus bioturbation and irrigation, as well as advection as a result of your stirring probably affects the flux.

**A: The diffusion-based fluxes are not overestimated at depth, where the resolution is sufficient and oxygen plays no role.**

Discussion
Line 334-7: Do you have data available to support that? Can you model that it does not fit?

**A: We have added two sentences in lines 547-553 to explain what a satisfying model would need to deliver. Such a model would have to be rather advanced and is not available to us, and would probably not be sufficiently constrained by the necessary physical background data.**

Line 342: How about rate studies in temperature gradient blocks, e.g. by Sagemann or Arnosti?

**A: We have unpublished temperature gradient block data on SRR for Himmerfjärden sediment. These support the existence of a broadly psychrotolerant/mesophilic SRB community and support the statement in the text. Apart from that, we would like to point out that temperature gradient block experiments of the kind the reviewer refers to are not adequate to address the temperature hypothesis for seasonal changes.**

Line 348: How big could the effect be with this difference in salinity?
**A: It is not relevant, mostly because sulfur cycling in the topmost cm makes sulfate multiple times available far in excess of concentrations variations due to salinity changes.**

Line 352: How much Corg comes from the sewage treatment plant?

**A: Although considerable (1676 tons for 2012, now mentioned in the text), the sewage treatment emits relatively small amounts of POM and DOM compared to the inorganic nutrients nitrate and phosphate, which stimulate plankton production in the estuary (Bonaglia et al., 2014). It is the nutrient effect that is most relevant for the carbon cycle in the estuary, not the heterotrophic carbon. Carbon estimates of the contribution by the sewage treatment plant were done by Savage et al. 2010 and indicate a local effect surrounding the sewage treatment plant due rapid deposition in the near area surrounding the outflow.**

How similar or different is the fjord thus to others?

**A: The inner parts of the fjord share similarities with other eutrophied fjord systems, e.g., Oslo fjord, whereas the outer parts are quite pristine and may be comparable to other northern latitude fjord-type systems. However, many fjord systems bordering the Atlantic have significantly higher salinities so that sulfate reduction prevails over a thicker sediment layer than in these sediments. Another difference is the glacial and postglacial history of the Baltic Sea. During glacial times organic-poor lake sediments were deposited followed Fe-rich post-glacial clay that is also low in organic carbon. In other fjord sediments, this discontinuity from a freshwater to a brackish/marine phase may not exist. This will affect the methane generation potential and thereby the methane flux.**

Line 357-9: Where do the high sedimentation rates come from if there is only low river runoff?

**A: These sediments are accumulation bottom sediments, which have a significant proportion of resuspended fine-grained material that is transported laterally and deposited in the bathymetric depressions of the fjärd.**

Line 385: Table 1 has no information about the burial of organic material. Do you have depth profiles supporting that?

**A: Published organic carbon concentration profiles can be found in Thang et al. (2013). Mass accumulation rates of organic carbon are reported in Thang et al. to be 9**

**– 9.5 mol m$^{-2}$ y$^{-1}$ (24 - 26 mmol m$^{-2}$ d$^{-1}$) for a station close to H6. At Station B1, we do not have information of mass accumulation. However, data for similar sediments suggest C$_{org}$ MAR of 3.3 mol m$^{-2}$ y$^{-1}$. We have added these data to the site description section.**

Line 391: Salinity of B1 is 7‰.

Line 392-4: A little too often "compar*"

**We have modified the text to address unnecessary repetitions.**

Line 392-8:
Lower sulphate concentrations mean that there is less sulphate available for organoclastic & methanotrophic sulphate reduction, just by simple numbers.

**We do not dispute this. This is stated in lines 428-432.**

Line 410&2: Figure 4

Line 412-3: Iron and manganese reducers do not always outcompete sulphate reducers, see work by Thamdrup and Vandieken.

**Yes, but in these sediments this is the case. We have conducted bag incubation and iron and Mn speciation analysis in bag incubations in core profiles to 10 cm depth. These data indicate that BSR account for 75% of anaerobic organic matter oxidation, Fe reduction about 6.5 % and Mn reduction about 2.5%. The rest is accounted for by heterotrophic denitrification (Goldschmidt presentation Downs and Bruchert (2013); Bonaglia et al. (2014) Biogeochemistry).**

Line 414-5: You state that the main driver for the differences is the advective flow based on the hydrostatic pressure, but here you speculate about more sulphate reduction leads to less methanogenesis. Which process is now the important one?

**A: W would like to emphasize that it is not one OR the other, but an interplay of multiple processes with varying influences on the system. This is also why these sediments would be extremely hard to model accurately in one-dimensional reaction-transport models.**

Line 417-20: How deep is the bioturbation in these sediments?
**A: An exact bioturbation depth would be arbitrary. Macrofauna analysis at H6 and B1 has shown that Marenzelleria does generally not go deeper than about 4 cm, but can occur occasionally down to 10 cm.**

Line 419-21: In advective systems with bioturbation fluxes should increase not decrease.

**A: We disagree. Since more oxygen can be imported, it is possible that methane oxidation increases.**

Line 443-5: This is not an explanation, it is just stating that you believe the data in contrast to the scenario below.

**A: Possible explanations are provided in the following lines. 479-487**

Line 454: "law" replace with "function"
**A: Done**

Line 461-3: Sentence not clear

**A: We rephrased the sentence: now lines 498-500**

Line 465: What is the percentage if you compare the methane flux into the SMTZ with the accumulated SRR or the total methane flux with the SRR? Do the numbers fit what model says?

**A: Lines 503 state what the reviewer asks for, i.e., the depth-integrated SRR relative to methane flux rates fit well with the model.**

Line 474-7: In line 445 you state that there is only little link.

**I am sorry but we do not see the apparent contradiction the reviewer states.**

Line 480-3: Please provide some support for this, maybe with a model.

**A: The data showed an abrupt decrease in porewater concentrations from Oct 2012 to January/February 2013 for which an explanation is required. The above discussion intended to lay out that neither temperature, salinity, nor changes in organic matter influx alone can explain this change. The sentence does not intend to say more than that. It is beyond the scope of the paper to develop a unifying model that can address these processes satisfactorily. Even some of the currently most complete models, such as by Mogollon et al. (2011) JGR are idealizations that may not yield satisfactory fits with our data, but that does not necessarily dispute either model or data.**

Line 484: Did you determine the concentrations of other electron acceptors like Fe? In line 319 you state that it is available for sulphide reoxidation.

**A: Fe data are available from the nearby Station H5 and published in Thang et al. (2013). In addition, there are an unpublished data for nearby Stations H3 and H2 that indicate the limitation of reactive iron in the postglacial mud. In the glacial lake clays, however, reactive iron is more abundant again, but these latter sediments do not control the methane production, because their $C_{org}$ contents are too low.**

Line 490-2: Do you have any data on changing pressures? What would drive these changes? What would the possible magnitude be?

**We have tried to obtain water level data and air pressure data for the periods of observation at the sampling stations, but these were not archived or could be found for the precise localities, only for the open Baltic nearby. However, local data is what was needed to have an accurate idea of the hydrostatic pressure.**
Line 500-1: Do you have any data or reference to support this magnitude?
**The best reference to address this question is the study by Mogollon et al (2011 JGR Biogeosciences), who modelled the free gas depth and AOM rates for two stations in southwestern Baltic Sea sediment. In that study temperature, and not tidally influenced pressure change, were found to be the dominant regulators of the free gas depth variation. Our differing intepretation is based on the observation that the seasonal variability in temperature at the two stites studied there are much greater than the ones studied here.**

Line 500: "may as much as: : :" insert "be"

**Done**
Line 497-503: If it is the hydrostatic pressure it is actually not really a seasonal effect?
**Yes**
Conclusion
Line 508-9: Sentence not clear.

Line 515-6: How is it seasonal/annual if it is the hydrostatic pressure? Not clear.

Line 517-9: You also state that you do not have data from the ice covered times!
**This must be a misunderstanding, since we have the February data. This is one of the few study that present data for conditions during ice cover.**
Line 519-20: Why is that if the temperature and Corg input do not play an important role?
**We emphasize that there are two aspects to be considered. Our winter and early spring data give low rates. Generally, in the literature, there are very few data available for sites with measurements during ice cover. If late spring/summer/fall rates are therefore extrapolated over a whole year, overestimates can result. Models, however, may account for this effect, if they are able to parameterize temperature and reactive organic carbon correctly (and the hydrological complexities of an archipelago setting riddled with fault lineaments). Secondly, while our observations indicate that hydrostatic changes or changes in porewater advection may have a considerable influence, this still does not take away from the fact that there is also seasonal variability. To model the annual variability based on organic carbon and temperature alone may therefore unfortunately also give the wrong results, because these factors have not been acounted to a degree that may reflect the specific regional situation. These two aspects need to be considered separately.**

Line 520-1: Is it no advective or diffusive transport that controls the methane? You keep changing your argument.
**Please see the comment above.**

Tables
Table 2: "no AOM zone3" for H6 January 2013, change to "no AOM zone4"

Table 3: "Exponential coefficient (a)" this one is not exponential.
**Will correct this**
Figures Make all the y axis for depth the same scale.
**Will correct this.**
Figure 1. The colors do not help reading the map too much, not very clear in black and white at all. Change continent to just white or black to make ocean more clear and maybe reduce shades in the water.
**We will consider a black and white map, or a map with sufficient contrast, clear legends and geographic locations.**
Figure 2: You use the maximum sulphate penetration from figure 3 here. Maybe change the order of the figure to keep the flow consistent.
**We will do that and then consequently also change the sequence in the results description.**
Figure 3: How do you define your maximum sulphate penetration if there is a sulphate peak in your graphs below it?
**Sulfate penetration depth was defined as the first lowest concentration measured. It is a common observation that traces of sulfate in the range of concentrations 50μM to 400 μM remain detectable in porewaters to substantial depths of several meters far into the methanogenic zone. This observation has been addressed in detail by Roey et al., Holmkvist et al and is part of the argument in favour of a cryptic sulfur cycle, which the authors of this manuscript also agree to. We also observe non-zero sulfate concentrations in the methane-rich zones of our sediment cores. We have therefore**

**chosen to define the sulfate penetration depth as that at which the sulfate changes sharply and the sulfate concentration is below 0.5 mM. We think that this combined criterion best reflects the condition of a transition from a dominating sulfate reduction environment to an environment in which methane-cycling processes start to prevail (while acknowledging low rates of cyptic bacterial sulfate reduction).**

Figure 5: Keep order consistent between listing in the graph and in the caption. If you keep all the values positive it is much easier to compare the values. Which method are the methane fluxes based on?

**We use the convention from the perspective of the water column, i.e., loss fluxes are fluxes into the sediment and negative and gain fluxes into the water column are positive.**

[revised manuscript text omitted]

---

## Referee Report (RR1)

2nd Review "Annual Variability and regulation of methane in sulphate fluxes in Baltic Sea estuarine sediments." by Joanna E. Sawicka and Volker Bruechert

Major comments:

Some of my previous comments have been answered in the response letter, but no appropriate changes have been made in the manuscript reflecting this answer (e.g. line 159: It still lists 2 hours drying time even thought the author answered that the actual drying time was longer; line 168: still says "exactly 2.5 mL" even though the author state it was changed; Line 170: the authors still do not cite a reference for fixation of the samples with 5M NaCl, even though they say it is a widely used method; Line 171: The authors agree that an hour is very likely does not get all the adsorbed methane. In the next answer, however, they state that the hour is actually sufficient and instead of clarifying that, the authors delete the information altogether and only state it in the response letter; Line 267: answering my comment, the authors state that they did some replication on methane profiles, but do not say anything about the fact or the results in the manuscript; Line 354: yes, there is the possibility of a cryptic sulphur cycle. But there is not a good reason to expect this cycle not to go on above. Thus, if you keep this information in there, you need to discuss it in more detail and add the information to the legend of the graph; line 367: please discuss the effect of the profile resolution on the modelling results...)

Unfortunately, the manuscript still in not organized all the way through with the same order for the stations in the results (e.g. line 300-304). Also, sometimes it seems that the authors are discussing a different data set (e.g. line 307: the graph shows the lowest sulphate penetration and steepest gradient in August; Line 318: Graph shows 16, 14, and 10cm (8cm in Figure 3); line 338: there is no sulphate in August below 10 or 8 cm, depending on the graph, so there can not be a peak in sulphate reduction, only in potential sulphate reduction)

The rates in table 2 are sometime negative and some positive. At the same time the direction they are described in change. A negative TOU means that there is oxygen diffusing out of the sediment. A positive $CH_4$ flux out of the sediment that there is methane diffusing out of the sediment, a negative $SO_4$ into the sediment means that there is sulphate diffusing out of the sediment. Please correct as mentioned before.

Line 486: you say that iron and manganese reducers in your sediment outcompete sulphate reducers (in your response, but again not in the manuscript, where you also do not mention that usually they do not). You cite Downs and Bruchert Goldschmidt 2013 and Bonaglia etal. 2014 Biogeochemistry (which both do not say anything about sulphate reduction). What do you base your statement on?

Comment to previous comment line 319: a 100-500um oxygen gradient from bottom water to 0 gives a high flux, which could be coupled to methane oxidation. Additionally, bioturbation can provide oxygen even below this depth.

Comment to previous comment line 465: Line 503 does not state what you mention in the answer, and I cannot find it. Additionally, it would be much better to get some numbers instead of just "fit well".

Comment to previous comment line 497-503: The authors agree that if it is hydrostatic than it is not seasonal. Additionally, they only have the data for one year, so it is hard to say which of the changes are clearly seasonal and which just variations over time. But they still argue for a seasonal signal.

Comment to previous comment line 519-520: You data does not "indicate that hydrostatic changes or changes in pore water advection may have a considerable influence". Your data indicates that the parameters you measured do not alone control the variations and you discuss, without presenting data or modelling, that it is hydrostatic changes and pore water advection.

Minor comments

Introduction

Line 60: remove "potentially"

Line 66: In the water methane can not be produced in the sediment, please rephrase.

Materials and Methods:

Line 113 and 423: you state that 23% of the freshwater is from a river, but also that there is no important river entering the system.

Line 134: where are the April 2013 data?

Line 155: what depth are the corg data from

Line 156/7 delete one mention of "freeze dried sediment"

Line 168: delete "exactly"

Line 170: now it sounds you measure the sample immediately after injection of the brine.

Line 218: switch the order

Line 242/3: move "without headspace", as now it sounds that the "50ul of 50% ZnCl2" did not have a a headspace.

Line 255: which salinity did you choose the ß for?

Results

Line 283: "sediment organic carbon"

Line 300ff: what is the trend in B1 and how is February B1 an exception to the H6 trend?

Line 307: the steepest gradient in the graph is shown in August.

Line 314: what is the trend in April and February?

Line 318: check the depths.

Line 322-34: state that August is much lower than April and October

Line 335: Keep the order the same as in the previous paragraph.

Line 338: Please state how you define the sulphate penetration depth as otherwise your rates below it do not make any sense.

Line 338: use the abbreviations you introduced e.g. SRR

Line 352: no change in sulphate concentrations argues for nor organoclastic just as much as for no AOM sulphate reduction.

Line 367: indicate which direction the methane flux is going.

Line 374: It is hard to discuss seasonal variation with 4 data points from one year.

Discussion

Line 383-391: Please link the statements directly to your data.

Line 397: there can also be imitations that occur at different temperatures, might be indirect though increased rates and thus increased competition, that can influence it. It does not only have to be a direct temperature effect.

Line 402-407: the argument is not convincing. For such small temperature differences, you do not expect to see changes in the membrane composition related to changes in temperature adaptation. And also closely related microbes can be psychrophiles and mesophiles and thus have a very different temperature

adaptation. Your data indicate that there is not such a tight coupling between sulphate reduction and methanogenesis.

Line 426: the percentage likely changes with season with resuspension being highest in fall, fresh organic matter being highest after a bloom and such. Thus arguing with general numbers for the whole year does not do justice to a potential effect of seasonality on the organic matter input. Also, your indicated bioturbation depth can supply fresh organic matter to deeper sediment depth, thus influencing rates not only on the surface.

Line 460 – 461: Please provide reference or calculation.

Line 485 – 488: that is not really what you see. The SRR below 10cm is lower than in other times, and thus, the effect on methanogenesis should be lower.

Line 491-494: First you argue that winter means higher bioturbation and higher (advective) methane flux, than you argue that winter means lower flux and concentration.

Line 517: What SRR can you sustain by the differences in the two CH4 fluxes? How does it compare to your rates?

Line 531: you ignore the second peak with this formular.

Line 542-544: Can you show the data.

Line 588-589: based on what? Please model to back up.

Tables

Table 2: how did you determine the AOM zone?

Figures

Map: the map still is not very helpful if plotted in black and white

Figures 2. 3. 5: Please delete the repeat mentioning of the months in the middle.

Figure 2: please provide legend showing which is CH4 and which is SO4. Change CH4 scale in (h) so the trend is more clear.

Figure 3: (a) and (b) seem to show a second peak around 8cm.

---

## Author Response (AR2)

2nd Response to reviewer "Annual Variability and regulation of methane in sulphate fluxes in Baltic Sea estuarine sediments." by Joanna E. Sawicka and Volker Bruechert

To the editor:

Dear Tina Treude,

We have now addressed the comments made by the reviewer and changed our manuscript accordingly. Please find all our changes listed below with line indications, where the changes have been made. The indicated lines refer to the document with the track changes. We have followed the reviewer's suggestions rather closely and agree with him on most points. We therefore feel that her/his thorough work has substantially improved the manuscript. Notably, the reviewer pointed to several inconsistencies between text, tables and figures that forced us to return to our original raw data. It turned out that some recalculations were necessary and these are now reflected in some of the new fluxes given in Table 2. It turned out that the discrepancy between the whole-core methane fluxes and the fluxes based on concentration gradients differed at both stations, which we acknowledge in the text. It is unfortunate that the whole-core fluxes do not agree better with the diffusive fluxes at the low-methane station, and this, as we state in the text, had possibly to do with some tough sampling conditions on a small vessel. However, we still feel that our discussion, which is mainly grounded in the porewater and rate data, is not affected by the methane whole-core flux data, and our overall conclusion holds. We would like to explicitly address the fact that we do not provide a fully integrated model of our data. We feel that even without the model our data are a significant scientific contribution. This study remains one of the few from this high latitude and in particular from the low-salinity regions of the Baltic Sea. It provides  substantial amount of new data for near-shore coastal sediment.

We hope that our revised version sufficiently satisfies the expectations of you and reviewer.

Sincerely and best regards,

Volker Brüchert and Joanna Sawicka

Lines refer to the version with track changes.

**Line 60: remove "potentially"**
A: removed

**Line 66: In the water methane cannot be produced in the sediment, please rephrase.**

A: The sentence now reads as follows: line 68
"In estuarine waters methane can be derived from underlying anoxic sediments, transported laterally due to freshwater or sewage discharge, seepage of methane-rich groundwater, or it can be derived from near-shore aquatic plants (Borges and Abril, 2011)."

**Line 113 and 423: you state that 23% of the freshwater is from a river, but also that there is no important river entering the system.**

A: 23% of total freshwater discharge came from the river Trosaån. However, this is still a very minor volume compared to the water exchange with the open Baltic. The average annual freshwater discharge given in Larsson (2013) is about 23 $m^3$ $s^{-1}$ from all freshwater sources and the total volume of the estuary given in Engqvist and Omstedt (1992) is 2968 x $10^6$ $m^3$. This translates into a turnover time give the freshwater discharge-based exchange time of 4 years. This is considerably slower than the exchange with the open Baltic (140 days bottom water and 70 days surface water) cited in Savage and Elmgren (2010) and indicated by the modest surface salinity gradient across the N-S section of Himmerfjärden.

Revised text: from line 113:
It is morphologically characterized by four basins, divided by sills and has a low flushing rate (~0.025/day) (Savage and Elmgren, 2010). The freshwater discharge is small compared to the exchange with the open Baltic and was estimated to be 23 $m^3$/s on average in 2012 comprising land run-off and precipitation ( 30% and 21% respectively), outflow from Lake Mälaren in the north (19%) and the river Trosaån (23%), and discharge from a sewage treatment plant (6%) (Larsson et al., 2012). The sewage treatment plant, built in the early 1970s, treats sewage water from ca. 314,000 inhabitants of the southern Stockholm metropolitan area, and its inorganic effluent is discharged mainly in the form of inorganic nitrogen and phosphorus to the inner basins (Savage and Elmgren, 2010). In 2012, the sewage treatment contributed 45% of the total phosphorus and 57% of the total inorganic nitrogen discharge to the northern Himmerfjärden area (Larsson et al., 2012) and discharged 1676 tons carbon (measured as chemical oxygen demand COD) (Stridh, 2012).

**Line 134: where are the April 2013 data?**

A: They are shown in Table 2.

**Line 155: what depth are the corg data from?**

A: Inserted " for the topmost cm of sediment"

**Line 156/7 delete one mention of "freeze dried sediment"**
A: Removed

**Line 159: It still lists 2 hours drying time even though the author answered that the actual drying time was longer;**
A: We do not have an exact record for how long the drying period exactly was, but the sediment was certainly dry. Time period has been removed from the text.

**Line 168: delete "exactly"**
A: Done

**Line 170: the authors still do not cite a reference for fixation of the samples with 5M NaCl, even though they say it is a widely used method;**

A: Added reference: Thang et al. (2013)

**Line 170: now it sounds you measure the sample immediately after injection of the brine.**

A: The text reads as follows:

A sediment sample of 2.5 mL was taken with a 3 mL cut-off syringe. The sample was transferred to a 20 mL serum vial containing 5 mL 5 M NaCl and immediately closed with a thick septum and an aluminum crimp seal (Thang et al., 2013). Before analysis, the sample was shaken and 5 mL of brine was injected into a sample vial to displace 5 mL gas out of a vial into the syringe. The $CH_4$ measurements were carried out on a gas chromatograph (GC) with a flame ionization detector (FID) (SRI 8610C) after separation on a 3 feet Porapak Q pre-column before a 9 feet Hayesep D column with $N_2$ as carrier gas.

**Line 171: The authors agree that an hour is very likely does not get all the adsorbed methane. In the next answer, however, they state that the hour is actually sufficient and instead of clarifying that, the authors delete the information altogether and only state it in the response letter;**

A: The reviewer misunderstood our comment. We do not think there is significant sorption after one hour. In any case, most of the analyses were performed later than one hour after sampling. The one hour period has been removed from the text.

**Line 218: switch the order**
Done (line 224)

**Line 242/3: move "without headspace", as now it sounds that the "50ul of 50% ZnCl2" did not have a a headspace.**
Done (line 248)

**Line 255: which salinity did you choose the ß for?**
**Added: "**using the salinity of the bottom water at the respective time point (Table 1)"

**Line 267: answering my comment, the authors state that they did some replication on methane profiles, but do not say anything about the fact or the results in the manuscript;**

A: See line 186: The reproducibility of the method has been tested at a station in the archipelago that is not part of this study by replicating methane sampling on multiple sediment cores. Concentrations were found to deviate by about 15%, likely due to shipment movement and sediment heterogeneity.

**Unfortunately, the manuscript still in not organized all the way through with the same order for the stations in the results (e.g. line 300‑‑‑304). Also, sometimes it seems that the authors are discussing a different data set (line 307).**
A: See changes in the text: lines
A: We certainly don't discuss another data set, but we apologize for the confusion. In any case, the correct SMT depths had been shown by the grey bars.

**Line 283: "sediment organic carbon"**
A: Changed

**Line 300ff: what is the trend in B1 and how is February B1 an exception to the H6 trend?**
The trends are now described in more detail in the text.

**Line 307: the graph shows the lowest sulphate penetration and steepest gradient in August;**
A: They are very similar in August and October at Station B1, but we agree that at Station H6 they are slightly steeper in August than in October. See changes in text line 310-327.

**Line 314: what is the trend in April and February?**
**A: Changed to: "**Generally, the sulfate gradient was similar throughout the measured profile in April and February, whereas in August and October, sulfate decreased more steeply from the surface down to 10 cm depth."

**Line 318: Graph shows 16, 14, and 10 cm (8cm in Figure 3);**
A: See changes lines 310-327. The correct sulfate penetration depths are now given for Station H6.

**line 319: a 100-500um oxygen gradient from bottom water to 0 gives a high flux, which could be coupled to methane oxidation. Additionally, bioturbation can provide oxygen even below this depth.**
A: Yes we agree, but we are a bit confused whether this comment corresponds to the line the reviewer refers to.

**Line 322-34: state that August is much lower than April and October**

**A: See changes in line 340**
Inserted the following sentence: "Contrary to expectations, the lowest sulfate reduction rates were measured in August."

**Line 335: Keep the order the same as in the previous paragraph.**
A: The section now reads: line 350
"At Station H6, depth-integrated sulfate reduction rates varied 9.2 to 11.7 mmol $m^{-2} d^{-1}$. The highest measured SRR was 338 nmol $cm^{-3} d^{-1}$ and occurred at 2 cm depth in April 2012. Organoclastic sulfate reduction dominated the interval down to 10 cm. In April, August, and October 2012 two distinct sulfate reduction rate peaks were found at station H6, one at the surface, and a second peak between 10 cm and 15 cm depth."

**Line 338: Please state how you define the sulphate penetration depth as otherwise your rates below it do not make any sense.**
We changed the term from sulphate penetration depth to 'minimum sulphate concentration depth' and sentence in line 329f of the previous section: "The initial depth at which sulfate reached the lowest concentration from the surface down was defined as the initial minimum sulfate concentration depth, which occurred at 16 cm depth in April, 10 cm in August, 14 cm in October and at 25 cm depth in February.".

**Line 338: use the abbreviations you introduced e.g. SRR**
A: We changed to SRR to be consistent.

**line 338: there is no sulphate in August below 10 or 8 cm, depending on the graph, so there cannot be a peak in sulphate reduction, only in potential sulphate reduction)**

A: This section of the text is a description of the data after addition of 1 mM sulfate with the tracer. As we commented on previously, the idea of adding 1 mM of sulfate was to avoid bias due to potential sulfate reduction. No addition of 'cold ' sulfate produces difficulties in comparing sulfate reduction rates across steep sulfate gradients. This problem has been addressed in previous publications (e.g., Holmkvist et al., 2014 GCA). We disagree with the reviewer to say there is 'no' sulfate below the SMT, because the ion chromatograph requires dilution of the sulfate sample (50 x dilution) (detection limit of the IC system 5µM), which raises the detection limit to a concentration of around 100 µM when calculating in the dilution. At this concentration there can be potentially sulfate reduction. We interpret the observed sulfate reduction increase in the SMT as a real peak in activity.

**The rates in table 2 are sometime negative and some positive. At the same time the direction they are described in change. A negative TOU means that there is oxygen diffusing out of the sediment. A positive CH4 flux out of the sediment that there is methane diffusing out of the sediment, a negative SO4 into the sediment means that there is sulphate diffusing out of the sediment. Please correct as mentioned before.**

A: We changed the TOU and sulfate reduction to positive values and the methane flux to negative values, and hope we have understood the reviewer correctly. We made sure there is a consistent use in text, table 2 and figure 4.

**Line 352: no change in sulphate concentrations argues for nor organoclastic just as much as for no AOM sulphate reduction.**
A: But the key here is that the methane concentrations do not change in gradient across this interval as would be expected if there was a significant coupling between the two due to AOM.

**Line 354: yes, there is the possibility of a cryptic sulphur cycle. But there is not a good reason to expect this cycle not to go on above. Thus, if you keep this information in there, you need to discuss it in more detail and add the information to the legend of the graph;**

A: We use the term potential sulfate reduction, because our data do not permit us to say otherwise. However, we would not like to expand beyond what we write here, just as much with a further discussion on potential cryptic sulfur cycling in shallower sediment layers. Short-term $^{35}$S incubations alleviate part of the cryptic sulfur cycling problem compared to gradient-based methods. We would also like to emphasize that a further discussion on cryptic sulfur cycling will not change our interpretation, which is primarily concerned with the flux of methane and the efficiency of AOM.

**line 367: please discuss the effect of the profile resolution on the modelling results…)**

A: In response to this we added lines 285-288 in section 2.10: "Since the resolution of the porewater methane analysis was 2 cm, steep concentration changes below this resolution cannot be captured. This could lead to an overestimation of the flux across the sediment surface, e.g., due to aerobic methane oxidation in the topmost mm of sediment. Similar effects may occur in the sulfate-methane transition zone."

**Line 367: indicate which direction the methane flux is going.**
A: Changed to "Upward diffusive fluxes …"

**Line 374: It is hard to discuss seasonal variation with 4 data points from one year.**
A: We agree, but there are distinct differences for the different time periods that fit with an
interpretation of seasonal variability.

**Line 383-391: Please link the statements directly to your data.**
A: line 386: Inserted: All R values calculated for pairs of temperature versus rate/flux were less than
0.2 …" and inserted "Table 1" at the end of line 391.

**Line 397: there can also be imitations that occur at different temperatures, might be**
**indirect though increased rates and thus increased competition, that can influence it. It**
**does not only have to be a direct temperature effect.**
A: Acknowledged. Since our data do not allow us to specify whether there are direct or indirect
effects at the organismal level, we restrict ourselves to the term methanogenesis and methane
oxidation
We changed the sentence, line 426-429: "This requires significantly higher temperature stimulation
of methanogenesis than methane oxidation, the lack of an electron acceptor, or competition for the
same electron acceptor used by other organisms than methane-oxidizing bacteria."

**Line 402-407: the argument is not convincing. For such small temperature**
**differences, you do not expect to see changes in the membrane composition related**
**to changes in temperature adaptation.**
   A.   We agree with the reviewer, but thisis exactly the  point we are making in the text.

**And also closely related microbes can be psychrophiles and mesophiles and thus**
**have a very different temperature adaptation. Your data indicate that there is not**
**such a tight coupling between sulphate reduction and methanogenesis.**
A: We think that the reviewer misinterpreted our argument. Our argument refers to the, as we
suggest, unlikely hypothesis that significant temperature adaptation differences should be
expected between Archaea involved in methanogenesis and Archaea involved in anaerobic
methane oxidation. However, as the reviewer wrote above, it is possible that the AOM process is
also not directly regulated by temperature adaptation of Archaea, and instead indirectly by a
temperature response of syntrophic SRB.
We have therefore inserted the following sentence, line 436-438: "However, temperature control may
not manifest itself by direct kinetic or bioenergetic regulation, but indirectly by influencing
competing processes, e.g., sulfate reduction versus methanogenesis. "

**Line 426: the percentage likely changes with season with resuspension being highest in**
**fall, fresh organic matter being highest after a bloom and such. Thus arguing with**
**general numbers for the whole year does not do justice to a potential effect of**
**seasonality on the organic matter input. Also, your indicated bioturbation depth can**
**supply fresh organic matter to deeper sediment depth, thus influencing rates not only on**
**the surface.**
A: The point that we are making is that the annual sedimentation only for a very short period of
time is dominated by primary organic carbon. We also agree with the notion that bioturbation
transports fresh organic matter to depth.
We modified the last sentence eof the paragraph as follows, line 468-474:

"Observations over a 5-year period by Blomqvist and Larsson (1994) indicated that primary organic carbon dominates organic sedimentation in the spring and summer at station B1, whereas station H6 is characterized by a spring term dominance of primary carbon deposition, but a much greater contribution of resuspended organic material to organic sedimentation (Blomqvist and Larsson, 1994)."

**Line 460 – 461: Please provide reference or calculation.**

A: The method is based on the assumption that organic matter reactivity can be described by a power law and follows a description initially proposed by Jorgensen (1979) and further used in Jorgensen and Parkes (2010) (cited). However, to our knowledge, the specific useage of this relationship for AOM superimposed on top of organoclastic sulfate reduction, is novel.

**Line 485 – 488: that is not really what you see. The SRR below 10cm is lower than in other times, and thus, the effect on methanogenesis should be lower.**

A: This would be true, if we ignored the methane concentration data and the sulfate penetration depths. However, our argument builds on the combined information.

**Line 486: you say that iron and manganese reducers in your sediment outcompete sulphate reducers (in your response, but again not in the manuscript, where you also do not mention that usually they do not). You cite Downs and Bruchert Goldschmidt 2013 and Bonaglia etal. 2014 Biogeochemistry (which both do not say anything about sulphate reduction). What do you base your statement on?**

A: In our response Bonaglia et al (2014) was used as reference for oxygen pentration depth, denitrification, anammox, and nitrate reduction with ammonium; Downs and Bruchert (2013) was used as reference for iron and manganese reduction. We don't have the combined information in one published reference, but in different papers. Since these studies were conducted at the same time, they can be used as reference.

We removed the following sentence from the text (line 527) : "Since the concentrations of other electron acceptors (e.g. for denitrification, iron, and manganese reduction) are highest in the topmost 3 cm (Bonaglia et al., 2014; Downs and Bruchert, 2013), the depth of sulfate penetration and organic matter degradation via sulfate shifts deeper in the sediment, which confines methane production to deeper layers."

**Comment to previous comment line 465: Line 503 does not state what you mention in the answer, and I cannot find it. Additionally, it would be much better to get some numbers instead of just "fit well".**

We are sorry, but our response referred to line 563, not line 503:

**Line 465: What is the percentage if you compare the methane flux into the SMTZ with the accumulated SRR or the total methane flux with the SRR? Do the numbers fit what model says?**

**Pleas see text (lines 589-594)**

"At station H6, between 5 % (August 2012) and 20% (April 2012) of the total sulfate reduction can be associated with anaerobic methane oxidation. A comparison of the above method with the integrated 35S-sulfate reduction rates integrated over the H6 sediment cores with the rates integrated over the AOM zone also indicated that >20% of sulfate respiration at H6 was fuelled by methane (Table 2).

**Comment to previous comment line 497---503: The authors agree that if it is hydrostatic than it is not seasonal. Additionally, they only have the data for one year, so it is hard to say which of the changes are clearly seasonal and which just variations over time. But they still argue for a seasonal signal.**
A. We agree that more data are needed to explore the effects of hydrostatic pressure changes, but this is what we write in the text.

**Comment to previous comment line 519---520: You data do not "indicate that hydrostatic changes or changes in pore water advection may have a considerable influence". Your data indicates that the parameters you measured do not alone control the variations and you discuss, without presenting data or modelling, that it is hydrostatic changes and pore water advection.**

A: We do not agree that we should not discuss the possibility that hydrostatic pressure changes and advection affect the methane concentration and fluxes. Our data should not be misunderstood that we favor one or the other hypothesis. But to leave our data without providing a possible scenario that could help to explain the variability does not seem correct. We are sorry but we cannot provide a encompassing model for the reasons presented in the previous discussion round. See our discussion lines 628-633 and 637-645.

**Copy from our first response letter:**
A: We emphasize that there are two aspects to be considered. Our winter and early spring data give low rates. Generally, in the literature, there are very few data available for sites with measurements during ice cover. If late spring/summer/fall rates are therefore extrapolated over a whole year, overestimates can result. Models, however, may account for this effect, if they are able to parameterize temperature and reactive organic carbon correctly (and the hydrological complexities of an archipelago setting riddled with fault lineaments). Secondly, while our observations indicate that hydrostatic changes or changes in porewater advection may have a considerable influence, this still does not take away from the fact that there is also seasonal variability. To model the annual variability based on organic carbon and temperature alone may therefore unfortunately also give the wrong results, because these factors have not been acounted to a degree that may reflect the specific regional situation. These two aspects need to be considered separately.

**Table 2: how did you determine the AOM zone?**
A: The AOM zone was defined a) from the overlap between sulfate and methane; b) an increase in sulfate reduction rates above the rates measured in the depth intervals above;

**Map: the map still is not very helpful if plotted in black and white**
A: We have changed the map one more time and also provide an insert of the greater Baltic Sea region.

**Figures 2. 3. 5: Please delete the repeat mentioning of the months in the middle.**
A: We removed the months.

**Figure 2: please provide legend showing which is CH4 and which is SO4. Change CH4 scale in (h) so the trend is more clear.**

A:We added a legend for sulfate and methane in in panel (h).  We would like to retain the same x-
scale as in the other panels for H6
**Figure 3: (a) and (b) seem to show a second peak around 8cm.**
Statistically this is difficult to support. Two of the injections showed higher rates, the third one
didn't.

[revised manuscript text omitted]

---

## Author Response (AR3)

3[rd] Response to manuscript "Annual Variability and regulation of methane in sulphate fluxes in Baltic Sea estuarine sediments." by Joanna E. Sawicka and Volker Bruechert

To the editor:

Dear Tina Treude,

We have now addressed your comments and made the recommended changes to the manuscript. Please find all our changes listed below with line indications of the present document, where the changes have been made.

Best regards,

Volker Brüchert and Joanna Sawicka (see lines : Please enter name of research vessel ("on-board" which vessel?)

633: Please provide a one sentence introduction of the purpose of this incubation at the beginning of the paragraph.
"In order to account for the total benthic exchange of oxygen and methane by advection, diffusion, bioirrigation, and bioturbation, four intact cores …"

and 653: I suggest to make these sub-capters of 2.7, because they belong to the incubation (2.7.1 and 2.7.2)
We changed the sections accordingly 2.7 w/ intro, then 2.7.1 for O2 uptake and 2.7.2 for **Benthic** methane fluxes

654: Methane fluxes of what? Benthic fluxes? Please add information.
Added **Benthic** methane fluxes

654: Was the water sample taken close to the sediment interface or in the bulk supernatant water. Please clarify.
added line 297: "directly above the sediment-water interface"

679: Chapter must than be 2.8 (see above)
Has been changed

735: Please comment that the detection limit of the IC is 100 µM for sulfate (see your comment in the answers to the reviewer)
Added section 2.5 line 248/249: The detection limit for sulfate after necessary dilution to reduce the chloride peak size was 100 µM.

744: Rates integrated over which depth?
Specified line 376 (B1) and line 389 (H6): "over the total core length"

865: "comparable" should better read "similar"
Changed in line 471

908: "Sediment focusing" sounds a bit strange. Rewording?

[revised manuscript text omitted]